# L$^2$M: Mutual Information Scaling Law for Long-Context Language Modeling

**Zhuo Chen**[12*]     **Oriol Mayné i Comas**[23]     **Zhuotao Jin**[24]
**Di Luo**[1245]     **Marin Soljačić**[12]

[1] NSF AI Institute for Artificial Intelligence and Fundamental Interactions
[2] Massachusetts Institute of Technology
[3] Polytechnic University of Catalonia
[4] Harvard University
[5] University of California, Los Angeles
{chenzhuo, omayne, jinzhta, diluo, soljacic}@mit.edu

## Abstract

We present a universal[†] theoretical framework for understanding *long-context language modeling* based on a *bipartite* mutual information scaling law that we rigorously verify in natural language. We demonstrate that bipartite mutual information captures multi-token interactions distinct from and scaling independently of conventional two-point mutual information, and show that this provides a more complete characterization of the dependencies needed for accurately modeling long sequences. Leveraging this scaling law, we formulate the **L**ong-context **L**anguage **M**odeling (**L$^2$M**) condition, which lower bounds the necessary scaling of a model's history state—the latent variables responsible for storing past information—for effective long-context modeling. We validate the framework and its predictions on transformer and state-space models. Our work provides a principled foundation to understand long-context modeling and to design more efficient architectures with stronger long-context capabilities, with potential applications beyond natural language.

## 1 Introduction

Large language models (LLMs) have revolutionized natural language processing, achieving remarkable capabilities across a wide range of tasks [1–4]. Recent advances in large language models, including ChatGPT [1, 5], Claude, Gemini [6, 7], Grok, LLaMA [4, 8], DeepSeek [9, 10], and Qwen [11, 12] have achieved breakthroughs across diverse tasks, including code generation, mathematical problem solving, text summarization, and creative writing [13–16]. These models have become increasingly powerful and versatile, pushing the boundaries of what's possible in natural language processing and marking significant steps toward artificial general intelligence [17–19].

In pushing these advances further, the ability to handle long contexts has become increasingly crucial. This ability is the key to document-level understanding, multi-turn dialogue, and complex reasoning. Models like GPT-o1/o3, Claude Opus, Gemini 2.5 pro, and DeepSeek-R1 often generate extensive chains of thought, spanning tens of thousands of tokens to solve complex problems [20, 21]. However, processing long contexts remains a significant challenge. Despite their success and expressiveness, transformer architectures suffer from an intrinsic quadratic computational cost in sequence length,

---

[*]Corresponding author

[†]Our framework applies to autoregressive language models, which encompass all widely-used LLMs such as GPT, Claude, Gemini, and LLaMA.

39th Conference on Neural Information Processing Systems (NeurIPS 2025).

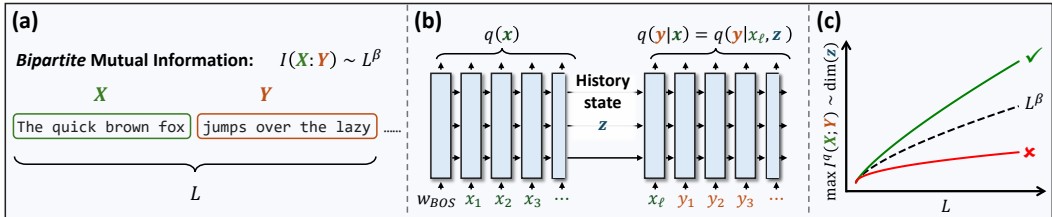

Figure 1: (a) The bipartite mutual information between two text segments scales as a power law (sub-volume law) with sequence length $L$. (b) In autoregressive models, conditional distributions are parameterized through the history state $z$, the latent variables that store past information. Examples of the history state include the recurrent states in state-space models or recurrent neural networks, and the key-value pairs in transformers. (c) The maximum bipartite mutual information a model can express scales with the dimensionality of its history state, $\dim(z)$. To model long contexts effectively, $\dim(z)$ must grow at least as fast as the power-law scaling of the true bipartite mutual information.

creating challenges for long sequence generation. Recent advances like DeepSeek have improved per-token efficiency [9], yet the fundamental quadratic cost persists.

Although various architectures have been proposed to address the quadratic scaling [22–31], these approaches still struggle with truly long sequences in practice. A fundamental gap persists in our theoretical understanding of what is necessary for capturing multi-token long-range dependencies in natural language. Despite efforts to characterize these dependencies through various statistical measures [32–35] a theory that can guide practical architecture design remains lacking.

In this work, we address the challenges of understanding long-context language modeling through the following contributions (Fig. 1).

1. We present a universal theoretical framework for autoregressive long-context language modeling based on bipartite mutual information.

2. We demonstrate a bipartite mutual information scaling law in natural language and provide reliable empirical validations of power-law scaling across diverse natural language datasets using state-of-the-art LLMs.

3. We derive the $L^2M$ condition from this scaling law, lower bounding the necessary scaling of a model's history state dimension for effective long-context modeling.

4. We validate our framework and its predictions across transformer and state-space model (SSM) architectures on both synthetic and natural language datasets of varying lengths.

Our theoretical framework offers crucial insights into understanding an LLM's capability to model long sequences based on its architectural design. By identifying the minimum required growth rate of the history state, our work provides concrete guidance for designing efficient architectures that can effectively handle long contexts, avoiding the quadratic cost of transformers or the capacity limitations of fixed-state models, paving the way for future AI systems.

## 2 Related Works

**Mutual Information Estimation and Application in Machine Learning**

Mutual information estimation and optimization have been extensively studied in machine learning, with approaches including variational bounds [36], neural estimators [37], nearest-neighbor methods [38, 39], and various upper bounds [40]. It has found wide application in areas such as feature selection [41], representation learning [42], disentanglement [43], and generative modeling [44].

**Statistical Properties of Natural Language**

Natural language exhibits characteristic statistical scaling behaviors across different levels of analysis. Zipf's law [45] describes how word frequencies decay with their rank, following a power-law distribution. Heaps' law [45] characterizes vocabulary growth, showing that the number of unique words scales sublinearly with text length. Hilberg's conjecture and its relaxed version posit specific scaling laws for entropy and bipartite mutual information in natural language, respectively [46].

### Neural Scaling Laws

Power-law relationships between model performance, architecture, and computational requirements have been first empirically observed in neural networks [47–49], with theoretical understanding still being developed [50, 51], including recent information-theoretic approach [52]. These observations have guided the development of larger models at fixed context lengths, whereas our work examines a distinct but complementary question: what determines whether model architectures can maintain performance as context length increases?

### Universal Prediction and Markov Modeling

Recent studies on transformers as universal predictors [53, 54] show that they can, in principle, model arbitrary variable-order Markov processes, establishing their theoretical universality in prediction. Our analysis focuses instead on the information-theoretic scaling that governs how much past information must be stored to reproduce the mutual-information growth observed in natural language.

### Architectures for Efficient Long-Context Modeling

Various approaches have been proposed for processing long sequences. Architectural innovations targeting quadratic complexity include sparse attention [55, 26, 25, 56], recurrent mechanisms [57–59], and alternative formulations [22, 60, 23, 27–30, 61–63, 31]. Efficient attention implementations like Flash Attention [64–66], Lightning Attention [67], and Paged Attention [68] have improved per-token computational efficiency despite maintaining the underlying complexity scaling.

### Long-Form Reasoning and Context Utilization

Chain-of-thought prompting [20] and scratchpad methods [69] demonstrate the importance of extended context for complex reasoning tasks, emphasizing the urgent need for effective long-range dependency modeling.

### Information Theory and Physics-Inspired Approaches

Recent work has demonstrated how information-theoretic principles and physics-inspired approaches can guide machine learning [70–72], leading to novel architectures [73–80], training methods [81–83], and broad applications [84–87, 82, 88].

## 3  Preliminaries

### Mutual Information

Mutual information $I(X;Y)$ quantifies the statistical dependence between random variables $X$ and $Y$, defined as $I(X;Y) = D_{KL}(p_{XY}||p_X \otimes p_Y)$, where $D_{KL}(\cdot||\cdot)$ is the Kullback–Leibler (KL) divergence, and $p_{XY}$ is the joint distribution of $X$ and $Y$. For discrete random variables, mutual information permits equivalent formulations as

$$I(X;Y) = H(X) + H(Y) - H(XY) = H(X) - H(X|Y) = H(Y) - H(Y|X), \tag{1}$$

where $H(\cdot)$ is the (Shannon) entropy and $H(\cdot|\cdot)$ is the conditional entropy. This definition naturally extends to collections of random variables: $I(X_{1:m}; Y_{1:n})$, with $X_{i:j}$ denoting the sequence $(X_i, \ldots, X_j)$. For notational convenience, we will use boldface notation $\boldsymbol{X} := X_{i:j}$ when the indices are clear from context. Similarly, we will drop the index of a single variable $X := X_i$ when convenient.

### Autoregressive Neural Networks

Modern LLMs predominantly employ autoregressive neural architectures. An autoregressive neural network models a sequence of conditional probability distributions over tokens $\{q(w_i|w_{1:i-1}, w_{\text{BOS}})\}_{i=1}^{L}$, where $w_{\text{BOS}}$ is the beginning-of-sequence token. Throughout this paper, we use $q$ to denote model-generated probability distributions (and sometimes the model itself) and $p$ to denote the true underlying distributions. Upper case letters denote random variables, and lower case letters denote specific values or realizations of these random variables. These conditional distributions jointly model the probability for a sequence of tokens given a prefix as

$$q(w_{\ell:L}|w_{1:\ell-1}, w_{\text{BOS}}) = \prod_{i=\ell}^{L} q(w_i|w_{1:i-1}, w_{\text{BOS}}). \tag{2}$$

When $\ell = 1$, this reduces to the distribution of unconditional generation $q(w_{1:L}|w_{\text{BOS}})$. During inference, tokens are sampled sequentially from these conditional distributions to generate text or respond to prompts.

For a complete list of notation and conventions used throughout this paper, we refer the reader to Appx. A.

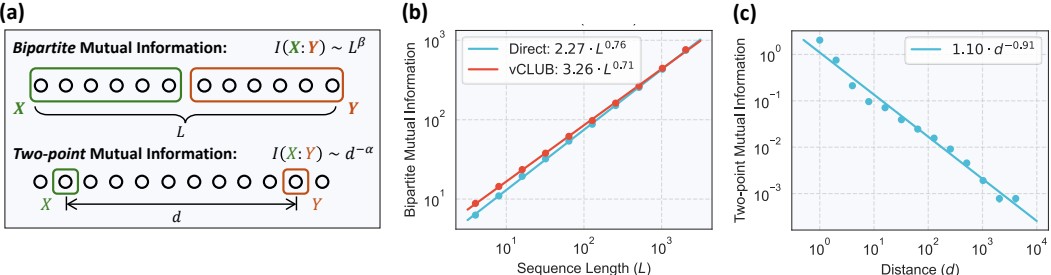

Figure 2: (a) Illustration of bipartite and two-point mutual information. The *bipartite* mutual information measures statistical dependence between two adjacent segments within a text block of length $L$, whereas the two-point mutual information measures the dependence between two tokens separated by a distance $d$. (b) Estimates of bipartite mutual information using LLaMA 3.1 405B model [89] on PG19 dataset [90] of pre-1919 books. (c) Estimates of two-point mutual information on PG19 dataset. See Appx. B.I, B.II, and B.VI for additional results.

## 4 Mutual Information Scaling Laws

### 4.1 Bipartite Mutual Information as Predictive Information

While classical scaling laws in natural language, such as Zipf's and Heaps' laws, primarily address token-level statistics, a deeper understanding of language modeling necessitates analyzing dependencies between entire text segments. A central challenge in modeling language effectively is to characterize how information is carried over from an existing block of text, $\boldsymbol{X}$, to inform the generation of a subsequent block, $\boldsymbol{Y}$. The *bipartite* mutual information between such adjacent blocks directly quantifies this inter-segment information transfer, emerging as a particularly revealing measure.

**Definition 4.1** (*Bipartite* Mutual Information [Fig. 2(a)]). For a consecutive sequence of tokens (random variables) $W_{1:L}$ of length $L$, consider a bipartition of the tokens: $X_{1:\ell} := W_{1:\ell}$ and $Y_{1:L-\ell} := W_{\ell+1:L}$. The bipartite mutual information is the mutual information between the two parts $I^{\text{BP}}_{\ell;L} := I(X_{1:\ell}; Y_{1:L-\ell})$.

The role of bipartite mutual information in quantifying this predictive relationship is formally illuminated by decomposing the entropy of the subsequent block $\boldsymbol{Y}$:

$$H(\boldsymbol{Y}) = H(\boldsymbol{Y}|\boldsymbol{X}) + I(\boldsymbol{X};\boldsymbol{Y}) = H(\boldsymbol{Y}|\boldsymbol{X}) + I^{\text{BP}}. \tag{3}$$

This decomposition shows that the total information in $\boldsymbol{Y}$ (its entropy $H(\boldsymbol{Y})$) consists of two distinct components: new information unique to $\boldsymbol{Y}$ given $\boldsymbol{X}$ (the conditional entropy $H(\boldsymbol{Y}|\boldsymbol{X})$), and information that $\boldsymbol{Y}$ shares with $\boldsymbol{X}$ (the bipartite mutual information $I^{\text{BP}}$). Consequently, $I^{\text{BP}}$ precisely measures the amount of information from the preceding block $\boldsymbol{X}$ that is predictive of the next block $\boldsymbol{Y}$, and therefore, bipartite mutual information is also referred to as predictive information [91].

Despite its crucial role in quantifying predictive information, this form of mutual information in language has remained relatively underexplored. This research gap is primarily due to two factors: the absence of a comprehensive theory of natural language that would permit a direct calculation, and the substantial challenges in empirically measuring entropy and mutual information for high-dimensional distributions from samples.

Existing literature offers differing perspectives on its scaling properties. On one hand, analogies drawing from critical physical systems [92–97]—often based on two-point mutual information

scaling (discussed later)—suggest that bipartite mutual information should scale logarithmically with sequence length. On the other hand, research in computational linguistics has proposed that it follows power-law growth [98], a behavior often referred to as the sub-volume law (these terms are used interchangeably in this paper). Previous empirical efforts to measure such scaling have been constrained by methodological biases and the curse of dimensionality [46, 98, 99]. Although existing evidence tends to favor sub-volume law growth, these limitations have prevented a definitive characterization. In Sec. 4.3, we address these challenges by leveraging state-of-the-art LLMs as density estimators, establishing clear power-law scaling for bipartite mutual information across diverse datasets.

## 4.2 Two-point Mutual Information

Before presenting our main results concerning bipartite mutual information scaling, it is instructive to discuss two-point mutual information. This measure has conventionally been used to assess long-range dependencies in natural language, and its scaling properties are relatively well understood.

**Definition 4.2** (*Two-point* Mutual Information [Fig. 2(a)])**.** The two-point mutual information measures the mutual information between two tokens (random variables) $X$ and $Y$ separated by a distance $d$: $I_d^{\mathrm{TP}} = I(X; Y)$.

Specifically, two-point mutual information has been observed to follow a power-law decay, $I_d^{\mathrm{TP}} \sim d^{-\alpha}$ [92–95]. This characteristic decay has prompted arguments that natural language shares structural properties with critical physical systems, which exhibit similar two-point correlation behavior [96, 97]. However, we contend that such analogies, while offering certain insights, can be misleading when assessing the full complexity of multi-token dependencies crucial for language modeling. The limitations of two-point mutual information in this regard, and why it provides an incomplete characterization for this task, will be detailed in Sec. 4.4 and Appx. B.VIII. Our present discussion of two-point mutual information serves primarily to contrast it with the bipartite measure that is central to our work.

## 4.3 Empirical Verification of Mutual Information Scaling Laws

**Bipartite Mutual Information.** Measuring bipartite mutual information presents significant challenges without access to the underlying probability distribution $p$. Traditional estimation methods face severe limitations in our setting: K-nearest neighbor estimators [39] and neural estimators like MINE [37] and InfoNCE [100] struggle with the high dimensionality of long text sequences, with errors that increase rapidly as sequence length grows. Additionally, neural estimators require substantial training on large amounts of data to learn representations of natural language distributions, especially for long sequences. Fortunately, recent advances in LLMs allow us to circumvent training our own density estimators by offering high-quality approximations $q$ to these distributions (see Appx. B.V for additional discussions). As autoregressive models, LLMs enable efficient computation of conditional probabilities (Sec. 3) and their associated cross-entropies (negative log-likelihoods):

$$H(p_{\boldsymbol{Y}|\boldsymbol{X}}, q_{\boldsymbol{Y}|\boldsymbol{X}}) := -\mathbb{E}_{p_{\boldsymbol{X}\boldsymbol{Y}}} \log q(\boldsymbol{Y}|\boldsymbol{X}), \tag{4}$$

where the expectation is taken over samples from the true underlying distribution $p_{\boldsymbol{X}\boldsymbol{Y}}$. The cross-entropy provides an upper bound to the true entropy:

$$H(p_{\boldsymbol{Y}|\boldsymbol{X}}, q_{\boldsymbol{Y}|\boldsymbol{X}}) = D_{KL}(p_{\boldsymbol{Y}|\boldsymbol{X}} \| q_{\boldsymbol{Y}|\boldsymbol{X}}) + H^p(\boldsymbol{Y}|\boldsymbol{X}) \geq H^p(\boldsymbol{Y}|\boldsymbol{X}), \tag{5}$$

where the conditional cross-entropy and KL divergence implicitly average over $p_{\boldsymbol{X}}$, and $H^p$ (or $H^q$) denotes the entropy computed with respect to distribution $p$ (or $q$).

Using these properties, we can construct a direct estimator for bipartite mutual information:

$$I_{\ell;L}^{\mathrm{BP,direct}} = \mathbb{E}_{p_{\boldsymbol{X}\boldsymbol{Y}}} \left[ \log q(\boldsymbol{Y}|\boldsymbol{X}) - \log q(\boldsymbol{Y}) \right] = I^p(\boldsymbol{X}; \boldsymbol{Y}) + \varepsilon(p, q), \tag{6}$$

where $I^p(\boldsymbol{X}; \boldsymbol{Y})$ denotes mutual information with respect to $p$, and $\varepsilon(p, q) = D_{KL}(p_{\boldsymbol{Y}} \| q_{\boldsymbol{Y}}) - D_{KL}(p_{\boldsymbol{Y}|\boldsymbol{X}} \| q_{\boldsymbol{Y}|\boldsymbol{X}})$. While this estimator no longer provides a bound, it preserves the key property that $\varepsilon(p, q) \to 0$ as $q \to p$.

We note that this estimation method faces a specific challenge with modern LLMs: they model $q(w_i | w_{1:i-1}, w_{BOS})$ rather than $q(w_i | w_{1:i-1})$, where $w_{BOS}$ denotes the BOS token. When sampling

from the dataset, we can ensure $\boldsymbol{X}$ starts at sentence beginnings, making $q(\boldsymbol{Y}|\boldsymbol{X}, w_{BOS}) \equiv q(\boldsymbol{Y}|\boldsymbol{X})$. However, $\boldsymbol{Y}$ may start mid-sentence, creating a mismatch where $q(\boldsymbol{Y}) \neq q(\boldsymbol{Y}|w_{BOS})$. This introduces errors in estimating $H(p_{\boldsymbol{Y}}, q_{\boldsymbol{Y}})$. We address this using $n$-gram corrections for the first two tokens, which are the primary source of this bias (see Appx. B.IV).

To circumvent issues with estimating $q(\boldsymbol{Y})$, we also employ the vCLUB estimator [40]:

$$I_{\ell;L}^{\mathrm{BP,vCLUB}} = \mathbb{E}_{p_{\boldsymbol{XY}}} \log q(\boldsymbol{Y}|\boldsymbol{X}) - \mathbb{E}_{p_{\boldsymbol{X}} \otimes p_{\boldsymbol{Y}}} \log q(\boldsymbol{Y}|\boldsymbol{X}), \tag{7}$$

where the second term can be calculated by shuffling the second halves of samples in the dataset. Analysis in [40] shows that vCLUB provides an upper bound on the true bipartite mutual information when $q$ closely approximates $p$. Even when $q$ deviates moderately from $p$, though the upper bound property may not hold, vCLUB continues to provide reliable estimates of the true bipartite mutual information.

Our empirical analysis in Fig. 2(b) focuses on equal-length partitions of $\boldsymbol{X}$ and $\boldsymbol{Y}$ ($\ell = L/2$), where the bipartite mutual information tends to maximize for fixed $L$'s. Nevertheless, the same analysis can be carried out using other partitions where similar results can be obtained (with the results in Appx. B.II). Using both the bias-corrected direct estimator [Eq.(6)] and vCLUB estimator [Eq. (7)], we measure scaling on the PG19 dataset* [90] (a collection of books before 1919), employing the LLaMA 3.1 405B model [89] as density estimator $q$. All measurements robustly demonstrate a clear power-law scaling that extends across thousands of tokens. Additional measurements on WIKIPEDIA [101] and using additional LLMs, along with varying $\ell/L$ ratios, can be found in Appx. B.I and B.II. We note that both estimators likely underestimate the true exponent $\beta$ (see Appx. B.III for discussions).

**Two-point Mutual Information.** For completeness, we also measure two-point mutual information scaling on the same datasets, confirming the expected power-law decay [Fig. 2 (c)]. Detailed methodologies for these measurements are provided in Appx. B.VI and B.VII.

## 4.4 Failures of Two-point Mutual Information

As previously noted, while two-point mutual information is easier to measure and more frequently studied in existing literature, it often fails to adequately capture the long-range multi-token dependencies crucial for natural language modeling. When modeling language, our primary concern is the accurate prediction of future tokens given a preceding context, i.e., $q(w_{\ell:L}|w_{1:\ell-1}, w_{BOS})$. Effective modeling of this conditional distribution necessitates a clear understanding of the multi-token dependencies between the context $w_{1:\ell-1}$ and the subsequent tokens $w_{\ell:L}$. It is important to recognize that this multi-token dependency cannot always be accurately represented by a simple aggregation of pairwise (two-point) interactions; such an approach can be insufficient or even misleading in certain contexts. The following examples illustrate these potential limitations, and we provide more formal derivations in Appx. B.VIII.

Consider a simple distribution where all tokens must be identical: $p(x_1, x_2, \ldots, x_L) = \mathbb{1}(x_1 = x_2 = \cdots = x_L)/M$, where $\mathbb{1}(\cdot)$ is the indicator function that evaluates to 1 when the condition is satisfied and 0 otherwise, and $M$ is the vocabulary size. This distribution permits a Markov chain construction, as $p(x_1) = 1/M$ and $p(x_i|x_{1:i-1}) = \mathbb{1}(x_i = x_{i-1})$, thus possessing a simple token-to-token dependency structure. Despite this inherent simplicity, the two-point mutual information suggests a misleadingly strong "long-range" dependency: it maintains a large, constant value of $I_d^{\mathrm{TP}} = \log M$ regardless of the distance $d$, significantly larger than the decaying two-point mutual information typically observed in natural languages. In contrast, bipartite mutual information correctly reflects this simple dependency structure, with $I_{\ell;L}^{\mathrm{BP}} = \log M$ remaining constant for any choice of $\ell$ and $L$. This indicates that any two segments share exactly the same amount of information ($\log M$), which is no more than the information shared between just two adjacent tokens, accurately capturing the limited nature of the dependency.

For a more realistic setting, we refer to Appx. C for a discussion of two families of multivariate Gaussian distributions of varying lengths (details of their construction are in Appx C.II). Notably, both families exhibit identical power-law decay in their two-point mutual information when measured between variables at maximum separation. However, their bipartite mutual information scaling differs dramatically: one scales as $L^\beta$, akin to natural language, while the other scales as $\log L$, similar to

---

*We avoid the BOOKS3 dataset due to copyright infringement concerns.

that observed in critical physical systems. This disparity further underscores that two-point mutual information alone may be insufficient to distinguish between systems with fundamentally different long-range correlational structures.

# 5 Long-Context Language Modeling (L$^2$M) Condition

Having established bipartite mutual information as a crucial tool for measuring long-range dependencies, we analyze how a model's capacity to handle long contexts fundamentally depends on its ability to store past information, using bipartite mutual information scaling as our theoretical framework. Intuitively, to model natural language effectively, a model must be able to capture all dependencies between past and future tokens. Since these dependencies (measured by bipartite mutual information) grow with sequence length, the model's state capacity for storing past information (the history state) must necessarily grow as well. We formalize this intuition through the L$^2$M condition and explore its implications in detail throughout this section.

## 5.1 Theoretical Derivations

To analyze how models handle long-range dependencies, we first formalize the notion of *history state*.

**Definition 5.1.** Consider a sequence of tokens $w_{1:L}$. Denote $x_{1:\ell} := w_{1:\ell}$ and $y_{1:L-\ell} := w_{\ell+1:L}$. Autoregressive neural networks parameterize conditional probabilities by first encoding the input tokens $x_{1:\ell-1}$ into a set of latent intermediate variables $\boldsymbol{z}_\ell = \boldsymbol{f}(x_{1:\ell-1})$ before outputting the conditional probabilities as $q(y_{1:L-\ell}|x_{1:\ell}) := q(y_{1:L-\ell}|x_\ell, \boldsymbol{z}_\ell)$.* We define the *history state* as the smallest set of such latent intermediate variables that fully characterizes the model's output conditional probability. [Fig. 1(b)].

As illuminating examples, the history state corresponds to the recurrent state in RNNs and SSMs after processing token $w_{\ell-1}$, and to the key-value pairs up to token $w_{\ell-1}$ for transformers (see Appx. D). Generally, the history state $\boldsymbol{z}_\ell$ is the smallest hidden state responsible for caching all historical information.

The following theorem shows that this history state upper bounds a model's capacity to capture bipartite mutual information:

**Theorem 5.2.** *The bipartite mutual information that a model can capture is bounded by the size of its history state:*

$$I^{\mathrm{BP},q}_{L/2;L} \leq C \cdot \dim(\boldsymbol{z}_{L/2}) + \log(M) \tag{8}$$

*where $C$ is some constant and $M$ denotes the vocabulary size.*

*Proof.* This theorem admits multiple independent proofs under different mild and practical assumptions. See Appx. E for details. □

We now use this bound to analyze when architectures can maintain performance as sequence length increases. Consider a series of natural language datasets $\{W_{1:L}\}_{L=1}^{\infty}$ of different lengths, which can be thought of as truncations of an ideal infinite-length dataset.

**Definition 5.3.** A model $q$ is *MI-capable* if the maximum bipartite mutual information it can express satisfies $\max_{\boldsymbol{\theta}} I^{\mathrm{BP},q_{\boldsymbol{\theta}}}_{L/2;L} \geq I^{\mathrm{BP}}_{L/2;L}$ for any sequence length $L$, where the maximum is taken over all model parameters $\boldsymbol{\theta}$.

Since a model's ability to capture mutual information is bounded by its history state dimension, we immediately obtain[†]:

---

*We separate $x_\ell$ from $\boldsymbol{z}_\ell$ to accurately reflect its distinct role as the current input token in autoregressive models, though including it in $\boldsymbol{z}_\ell$ would not affect the main results of this paper.

[†]See Appx. A for conventions on asymptotic notations.

> **Theorem 5.4** (L$^2$M Condition for Single Models). *For a model to be* MI-capable *across all sequence lengths, its history states $\boldsymbol{z}_{L/2}^q$ must satisfy* $\dim(\boldsymbol{z}_{L/2}^q) \succsim I_{L/2;L}^{\mathrm{BP}} \sim L^{\beta}$.

*Proof.* We prove by contrapositive. By Thm. 5.2, if $\dim(\boldsymbol{z}_{L/2}) \prec I_{L/2;L}^{\mathrm{BP}}$, then $\max_{\boldsymbol{\theta}} I_{L/2;L}^{\mathrm{BP},q_{\boldsymbol{\theta}}} \prec I_{L/2;L}^{\mathrm{BP}}$, implying there exists some $L$ where $\max_{\boldsymbol{\theta}} I_{L/2;L}^{\mathrm{BP},q_{\boldsymbol{\theta}}} < I_{L/2;L}^{\mathrm{BP}}$, violating MI-capability. $\qquad\square$

For some architectures, a single fixed-size model may not satisfy this condition across all sequence lengths. In such cases, we can extend our framework to families of models where model size grows with sequence length. Consider a series of models $\{q_L\}_{L=1}^{\infty}$ of the same architecture, where model size may increase with $L$.

**Definition 5.5.** A series of models $\{q_L\}_{L=1}^{\infty}$ is *MI-capable* if the maximum bipartite mutual information each model can express satisfies $\max_{\boldsymbol{\theta}_L} I_{L/2;L}^{\mathrm{BP},q_L,\boldsymbol{\theta}_L} \geq I_{L/2;L}^{\mathrm{BP}}$ for its corresponding sequence length $L$, where the maximum is taken over all parameters $\boldsymbol{\theta}_L$ of model $q_L$.

> **Theorem 5.6** (L$^2$M Condition for Model Series). *For a series of models $\{q_L\}_{L=1}^{\infty}$ to be* MI-capable, *the history states $\boldsymbol{z}_{L/2}^{q_L}$ of each model must satisfy:* $\dim(\boldsymbol{z}_{L/2}^{q_L}) \succsim I_{L/2;L}^{\mathrm{BP}} \sim L^{\beta}$.

Note that an MI-capable single model trivially induces an MI-capable series when applied to all sequence lengths, though the converse is not true.

## 5.2 Implications to Common LLM Architectures

We can now apply our framework to analyze whether different architectures satisfy the L$^2$M condition and thus can capture long-range dependencies as sequence length grows.

In transformer-based models (excluding sparse attention and linear attention variants), the history state consists of stored key-value pairs for all previous tokens. Even with fixed model size, these key-value pairs grow linearly with sequence length: $\dim(\boldsymbol{z}_{L/2}^q) \sim L \succsim L^{\beta}$. This means a single transformer model naturally satisfies the L$^2$M (single model) condition across all sequence lengths, notwithstanding the quadratic computational cost.

In contrast, SSMs, RNNs, and linear attention models, despite being celebrated for their "infinite" context length and linear complexity, cannot satisfy the L$^2$M condition with a single fixed-size model. Their history state dimension remains constant regardless of sequence length, and our theory demonstrates that this constant-size state cannot capture the growing mutual information. However, these architectures can achieve MI-capability (model-series) through a series of models $\{q_L\}_{L=1}^{\infty}$ where model size, and thus history state dimension, increases with sequence length. This requirement effectively offsets their computational efficiency advantage when modeling long sequences.[*]

For other architectures, such as sparse attention models and log-linear models, we can similarly analyze their history state scaling to determine whether they satisfy the L$^2$M condition as single models or require a series of growing models. Crucially, any architecture must exhibit power-law growth in its history state dimension with sequence length in order to truly satisfy the single-model L$^2$M condition.

We note that the L$^2$M condition addresses a model's capacity to capture long-range dependencies, not its overall language modeling capability. It is a necessary but not sufficient condition: architectures that fail to satisfy it will have inherent limitations at longer sequences, while satisfying it does not guarantee effective language modeling. As discussed in Sec. 2, the L$^2$M condition is also distinct from neural scaling laws, which typically study how model performance scales with model size, dataset size, and compute budget at a *fixed* sequence length.

---

[*]And a new model must be trained for each sequence length, which can be prohibitively expensive.

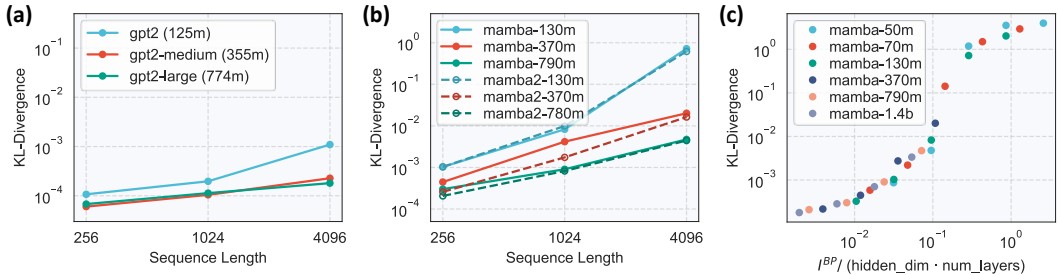

Figure 3: Evaluation of KL-divergence across model architectures trained on synthetic data that satisifes the bipartite mutual information scaling. (a, b) Average KL-divergence per token for models trained on different sequence lengths. (c) Average KL-divergence per token as a function of the ratio between bipartite mutual information and Mamba recurrent state sizes.

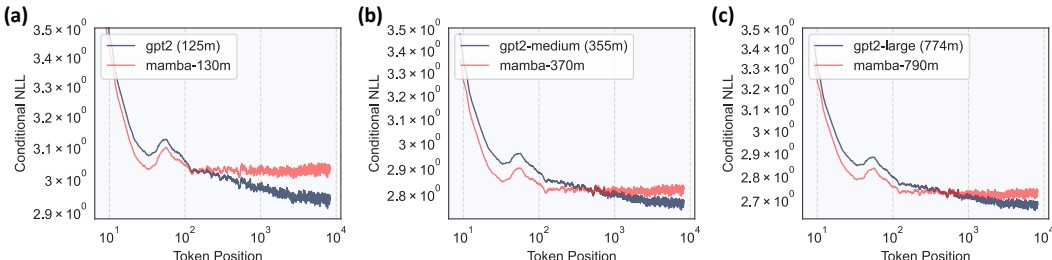

Figure 4: Position-wise conditional negative log likelihood (NLL) evaluation for models trained on 8192-token sequences on the PG19 dataset [90].

## 6 Empirical Verification

**Sub-volume-law Gaussian Test.** We first validate our theory using a synthetic dataset comprising a family of multivariate Gaussian distributions (see Appx. C for details). This distribution family closely mimics the scaling of both bipartite and two-point mutual information observed in natural language, while crucially allowing for the efficient calculation of conditional probabilities and KL divergences—calculations that would be intractable with real-world natural language datasets. Furthermore, the synthetic data enables an isolated assessment of a model's ability to handle long sequences, without interference from its capacity to understand the semantic meanings of natural language.

In Fig. 3, we present the average per-token KL divergence (defined in Appx. F.IV) for GPT2, Mamba, and Mamba2 models, serving as representative transformer and SSM architectures. Panels (a) and (b) show that GPT2 maintains consistent KL divergence across different sequence lengths. In contrast, smaller Mamba and Mamba2 models exhibit increasing difficulty with longer contexts, necessitating substantially larger model sizes to achieve comparable performance at a sequence length of 4096. Panel (c) offers direct confirmation of our theoretical framework: it plots KL divergence against the ratio of bipartite mutual information to the recurrent state size for Mamba models of varying configurations. For this, we varied sequence lengths from 64 to 16,384 and model sizes from 50M to 1.4B parameters. The KL divergence values from these diverse configurations remarkably collapse onto a single curve, demonstrating that model performance depends only on the ratio $I^{\mathrm{BP}}/\dim(\boldsymbol{z})$. This finding precisely confirms our theory that for effective long-context modeling, a model's history state size must scale at least as fast as the bipartite mutual information present in the data.

These findings have important implications for modeling very long sequences. Extrapolating from the measured scaling in Fig. 2 (which likely underestimates the true exponent), the bipartite mutual information for a sequence of one million tokens could exceed 60,000 nats. Our results in Fig. 3(c) suggest that maintaining low KL divergence at such bipartite mutual information levels would require recurrent state dimensions approaching one million.

**PG19 Test.** We then extend our analysis to the PG19 dataset [90], a high-quality collection of pre-1919 books exhibiting long contextual dependencies.

In Fig. 4, we show the position-wise conditional negative log likelihood (NLL) of models trained on the PG19 dataset [90] with 8192-token sequences, where calculating KL-divergence is not feasible.

Note that, unlike conditional KL divergence, conditional NLL naturally decreases with token position (see Appx. F for details). Two key patterns emerge from this experiment: First, Mamba models typically outperform GPT2 models of comparable size at early token positions, but this advantage diminishes and eventually reverses at later positions. Most notably, Mamba's NLL tends to plateau beyond certain positions unless the model size is increased, while GPT2's NLL continues to improve. Second, the performance gap between Mamba and GPT2 narrows with increasing model size. Both observations align with our theoretical predictions: since Mamba's history state size remains fixed regardless of sequence position, its performance inevitably degrades beyond a certain token position unless model size increases. As model size grows, the history state size also increases, eventually becoming sufficient to capture the mutual information present in 8192-token sequences.

We note that Mamba's linear computational complexity can make larger Mamba models practically more efficient than smaller transformers. Our results should not be interpreted as suggesting Mamba's architectural inferiority. Rather, they demonstrate how different architectures handle long sequences differently, and that a model's capacity for capturing long-range dependencies aligns with our theoretical $L^2M$ framework, regardless of the architecture.

Additional experimental results can be found in Appx. G.

# 7 Discussion

The $L^2M$ condition establishes a fundamental relationship between the information structure of data and architectural requirements. This relationship manifests differently across architectures: transformers with linearly growing key-value caches naturally satisfy the condition as single models (given our measured sublinear mutual information scaling with $\beta < 1$), though at quadratic computational cost, while SSMs and similar fixed-state architectures require model size to scale with sequence length to achieve comparable mutual information capability.

Interestingly, transformers appear to *over-provision* their history state relative to the measured mutual information scaling: their linear growth exceeds the sublinear ($L^\beta$ with $\beta < 1$) scaling we observe. This observation provides a clear goal for future architecture design. Although it remains unclear whether the over-provisioning is necessary for other aspects of language modeling beyond pure information storage, the gap between the linear growth of transformers and the $L^\beta$ requirement suggests a concrete target: architectures that precisely match the required sublinear scaling could potentially achieve substantially improved efficiency while maintaining the capacity to capture long-range dependencies.

Our framework applies to autoregressive language models, which encompass the vast majority of widely-used LLMs. While diffusion-based language models represent an alternative generative paradigm, they typically still operate autoregressively at a higher level of granularity, making our framework applicable in practice. Extending our framework to hybrid architectures that combine different mechanisms represents an important research direction that could unify our understanding of how diverse architectural choices affect long-context capabilities. Applying our framework to other sequential domains, such as biological sequences like proteins or DNA, or computer code, also presents a particularly promising direction, as different mutual information scaling behaviors in these domains could provide a principled explanation for the observed differences in model requirements across domains.

Additional discussions on limitations and broader impacts can be found in Appx. H and I.

# 8 Conclusion

We establish a bipartite mutual information scaling law that characterizes long-range dependencies in natural language and introduce the $L^2M$ condition, which lower bounds the necessary scaling of a model's history state for effective long-context modeling. By identifying the minimum required growth rate of the history state, our work provides a principled foundation for understanding how different architectures handle long contexts. This framework establishes concrete, information-theoretical, and data-driven targets that could guide the design of architectures balancing computational efficiency with the capacity to capture long-range dependencies in natural language and potentially beyond.

## Acknowledgements

The authors acknowledge support from the National Science Foundation under Cooperative Agreement PHY-2019786 (The NSF AI Institute for Artificial Intelligence and Fundamental Interactions). Z.C. acknowledges support from the Mathworks Fellowship. Z.C. thanks Rumen Dangovski for helpful discussions. Z.C. and O.M. thank Amazon Web Services account team, including Brian McCarthy and Jared Novotny, for technical support. The research was sponsored by the United States Air Force Research Laboratory and the Department of the Air Force Artificial Intelligence Accelerator and was accomplished under Cooperative Agreement Number FA8750-19-2-1000. The computations in this paper were partly run on the FASRC cluster supported by the FAS Division of Science Research Computing Group at Harvard University. This research used the DeltaAI advanced computing and data resource, which is supported by the National Science Foundation (award OAC 2320345) and the State of Illinois, through allocation CIS240904 from the Advanced Cyberinfrastructure Coordination Ecosystem: Services & Support (ACCESS) program, supported by National Science Foundation grants #2138259, #2138286, #2138307, #2137603, and #2138296, and through the National Artificial Intelligence Research Resource (NAIRR) Pilot NAIRR250043.

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

# A   Notations and Conventions

**Basic notations**

- Random variables: Uppercase letters (e.g., $X$, $Y$, $W$) denote random variables.
- Realizations: Lowercase letters (e.g., $x$, $y$, $w$) denote specific values or realizations of random variables.
- Sequences: $X_{i:j}$ denotes the sequence $(X_i, X_{i+1}, \ldots, X_j)$.
- Bold notation: $\mathbf{X} := X_{i:j}$ when indices are clear from context.
- Single variable shorthand: $X := X_i$ when the index is clear from context.
- Logarithms: While the choice of base does not affect the scaling laws (only the multiplicative constants), all logarithms are natural logarithms (base $e$) unless otherwise specified.

**Information-theoretic quantities**

- $H(\cdot)$: Shannon entropy.
- $H(\cdot|\cdot)$: Conditional (Shannon) entropy.
- $I(\cdot;\cdot)$: (Shannon) mutual information.
- $D_{\mathrm{KL}}(\cdot||\cdot)$: Kullback–Leibler divergence.
- $H^p(\cdot)$: Entropy computed with respect to distribution $p$.
- $H^q(\cdot)$: Entropy computed with respect to distribution $q$.
- $H(p, q)$: Cross-entropy between distributions $p$ and $q$.

**Asymptotic notations**

- $f(n) \sim g(n)$: $f$ and $g$ have the same asymptotic growth rate, i.e., $f(n) = \Theta(g(n))$.
- $f(n) \succ g(n)$: $f$ grows strictly faster than $g$ asymptotically, i.e., $f(n) = \omega(g(n))$.
- $f(n) \succsim g(n)$: $f$ grows at least as fast as $g$. asymptotically, i.e., $f(n) = \Omega(g(n))$.
- $f(n) \prec g(n)$: $f$ grows strictly slower than $g$ asymptotically, i.e., $f(n) = o(g(n))$.
- $f(n) \precsim g(n)$: $f$ grows at most as fast as $g$ asymptotically, i.e., $f(n) = O(g(n))$.

**Distributions and expectations**

- $p$: True underlying probability distribution (of natural language).
- $q$: Model-generated probability distribution (sometimes refers to the model itself).
- $\mathbb{E}_p[\cdot]$: Expectation with respect to distribution $p$.
- $p_X \otimes p_Y$: Product distribution of marginals $p_X$ and $p_Y$.

**Model-specific notations**

- $w_{\mathrm{BOS}}$: Beginning-of-sequence token.
- $M$: Vocabulary size.
- $L$: Sequence length.
- $\ell$: Position of sequence split for bipartite mutual information.
- $\dim(\boldsymbol{z})$: Dimensionality of the history state $\boldsymbol{z}$.
- $\theta$: Model parameters.

**Special notations**

- $I^{\mathrm{BP}}$: Bipartite mutual information.
- $I^{\mathrm{TP}}$: Two-point mutual information.
- $\mathbb{1}(\cdot)$: Indicator function (equals 1 when condition is true, 0 otherwise).

The notations are used consistently throughout the main text and appendices unless otherwise specified in local contexts.

# B  Additional Details on Mutual Information Scalings

## B.I  Bipartite Mutual Information Scaling with Additional LLMs on Additional Datasets

In the main text, we use the LLaMA 3.1 405B model as the density estimator and measured the bipartite mutual information scaling on PG19 dataset. In this section, we provide additional estimations of the bipartite mutual information scaling using the DeepSeek V3 Base model and on WIKIPEDIA dataset. We note that because we are merely measuring the conditional probabilities of the input tokens without interactions with the agent, we believe the non-instruction-finetuned model better suits our tasks.

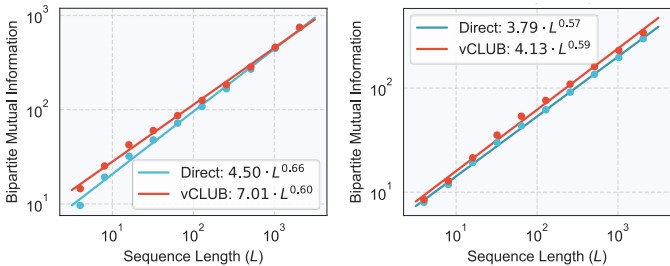

Figure B.1: Bipartite mutual information estimation using (left) LLaMA 3.1 405B on the WIKIPEDIA dataset and (right) Deepseek V3 Base model on the PG19 dataset. All direct measurements include the bias correction described in Appx. B.IV.

In Fig. B.1, we report the results on WIKIPEDIA dataset using LLaMA 3.1 405B model as the density estimator and on PG19 dataset using Deepseek V3 Base model as the density estimator. We find that in both cases, clear sub-volume growth behavior is observed. We note that the measured exponent should be taken with a grain of salt and likely underestimates the true mutual information scaling due to reasons explained in Appx. B.III.

## B.II  Bipartite Mutual Information Scaling Under Various Ratios of $\ell/L$

In the main text, we focused on the bipartite mutual information with equal splits. However, the bipartite mutual information scaling is not limited to equal biparitition. In this section, we provide additional results for various ratios of $\ell/L$.

In Fig. B.2, we provide estimation of the bipartite mutual information scaling for $\ell/L = 3$ and $\ell/L = 4$. All results show clear power-law relations, and are consistent with Fig. 2 in the main text. These results can be used to support the L$^2$M condition with similar arguments as in the main text.

## B.III  Why The Estimated Exponent $\beta$ Is Likely An Underestimation?

In the main text, we mentioned that our measured exponent $\beta$ using LLMs likely underestimates the true $\beta$. Here, we discuss the reasons.

For the direct estimator,

$$I_{\ell;L}^{\mathrm{BP,direct}} = H(p_{\boldsymbol{Y}}, q_{\boldsymbol{Y}}) - H(p_{\boldsymbol{Y}|\boldsymbol{X}}, q_{\boldsymbol{Y}|\boldsymbol{X}}), \tag{B.1}$$

both terms (without the minus sign) overestimates the true (conditional) entropy, but for different extent and at different scales.

At small $L$, the first term suffers from the bias from the BOS token as discussed in Appx. B.IV. The second term, despite also an overestimation, does not suffer from the BOS token issue. Therefore, at small $L$, the direct estimator tends to overestimate the true entropy.

At large $L$, the bias from the BOS token is less severe. However, modeling $p(\boldsymbol{Y}|\boldsymbol{X})$ requires the model to correctly capture all the dependencies between $\boldsymbol{X}$ and $\boldsymbol{Y}$, making it significantly harder than modeling $p(\boldsymbol{Y})$ alone. Therefore, $q(\boldsymbol{Y}|\boldsymbol{X})$ is likely a worse estimation of the true distribution than $q(\boldsymbol{Y})$, resulting in more overestimation in the second term, and an underestimation of the bipartite mutual information.

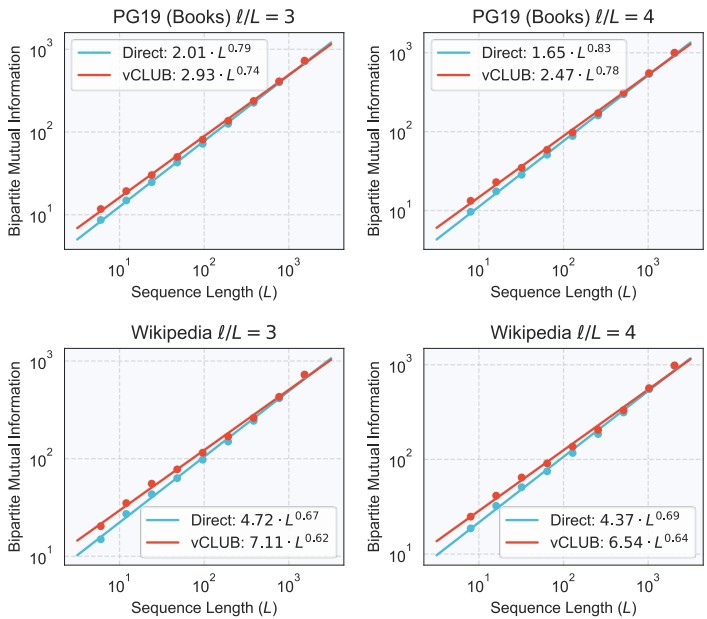

Figure B.2: Bipartite mutual information estimation using different ratios of $\ell/L$. All results suggest the existence of power-law scaling, with various fitted exponents.

This means that the direct estimator tends to overestimate the true bipartite mutual information at small $L$ and underestimate it at large $L$, resulting in an underestimation of the fitted exponent.

The vCLUB estimator, as pointed out in [40], is an upper bound to the true mutual information if $q$ is close to $p$, but fails to maintain the property when the KL-divergence between them increases. Therefore, it is likely that this estimator also overestimates the true bipartite mutual information at small $L$ and underestimates it at large $L$, resulting in a similar underestimation of the fitted exponent as our direct estimator. As our fitted exponent for the vCLUB estimator is smaller than that of the direct estimator, we conclude that the vCLUB estimator has a larger bias in this case, and it is reasonable to believe that the true exponent is even larger.

### B.IV  Direct Estimation of Bipartite Mutual Information Using LLMs

In the main text, our direct estimator for the bipartite mutual information is

$$I_{\ell;L}^{\text{BP,direct}} = \mathbb{E}_{p_{\boldsymbol{XY}}}\left[\log q(\boldsymbol{Y}|\boldsymbol{X}) - \log q(\boldsymbol{Y})\right] = H(p_{\boldsymbol{Y}}, q_{\boldsymbol{Y}}) - H(p_{\boldsymbol{Y}|\boldsymbol{X}}, q_{\boldsymbol{Y}|\boldsymbol{X}}) = I^p(\boldsymbol{X};\boldsymbol{Y}) + \varepsilon(p,q). \tag{B.2}$$

where as usual, $\boldsymbol{X} := X_{1:\ell} := W_{1:\ell}$ and $\boldsymbol{Y} := Y_{1:L-\ell} := W_{\ell+1:L}$ with $W_{1:L}$ being a sequence of tokens. However, as discussed in the main text, the $H(p_{\boldsymbol{Y}}, q_{\boldsymbol{Y}})$ term suffers from an additional bias—we cannot guarantee that $\boldsymbol{Y}$ starts at the beginning of a sentence, but LLMs model distributions conditioned on BOS token. To mitigate this issue, we use $n$-gram calculations to correct the entropy of the first two tokens as explained below.

We first rewrite the (marginal) cross entropy as

$$H(p_{\boldsymbol{Y}}, q_{\boldsymbol{Y}}) = -\mathbb{E}_p[\log q(\boldsymbol{Y})] = -\sum_{i=1}^{L-\ell} \mathbb{E}_p[\log q(Y_i|Y_{1:i-1})], \tag{B.3}$$

where as usual, the expectation over the conditional variable is omitted but implied in the cross entropy calculation.

In modern LLMs, we can only compute $q(y_i|y_{1:i-1}, w_{BOS}) \neq q(y_i|y_{1:i-1})$, resulting in an additional error in the bipartite mutual information estimation. In practice, this difference becomes less pronounced for larger $i$, because it matters less if the sequence starts at the beginning of a sequence or not if there are many $y_{1:i-1}$ prior tokens to conditional on. Therefore, we focusing on reducing the

bias for small $i$. In addition, if $i$ is small, we can iterate over the dataset and construct a histogram for the $i$-gram distribution $p(y_{1:i})$.

We denote the count for each $i$-tuple of tokens with $n_{y_{1:i}}$ and the total number of samples with $N$. Then, the entropy of the distribution can be estimated naively as

$$\hat{H}^{\text{naïve}}(Y_{1:i}) = -\sum_{y_{1:i}} \frac{n_{y_{1:i}}}{N} \log \frac{n_{y_{1:i}}}{N} = \log N - \frac{1}{N} \sum_{y_{1:i}} n_{y_{1:i}} \log n_{y_{1:i}}, \tag{B.4}$$

where the summation runs over all possible combination of tokens $y_{1:i} := (y_1, y_2, \ldots y_i)$.

However, this estimation is severely biased and underestimates the true entropy, due to the concavity of logarithm function. In [102], a bias-corrected estimator is proposed by replacing the logarithm function with a new function

$$\hat{H}^G(Y_{1:i}) = \log N - \frac{1}{N} \sum_{y_{1:i}} n_{y_{1:i}} G(n_{y_{1:i}}), \tag{B.5}$$

where

$$G(n) = \psi(n) + \frac{(-1)^n}{2} \left( \psi\left(\frac{n+1}{2}\right) - \psi\left(\frac{n}{2}\right) \right), \tag{B.6}$$

with $\psi(\cdot)$ the digamma function. We note that Ref. [102] was not able to obtain the closed form expression for $G(\cdot)$, which we derived with the help of Wolfram Mathematica [103].

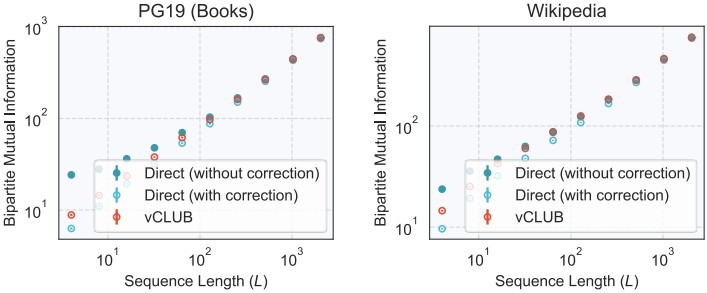

Figure B.3: Effect of bias correction method in the direct estimator. The bias only affects the estimation at small sequence lengths, and all methods converge at large sequence lengths.

This bias-corrected estimator still underestimates the true entropy, but much less compared to the original naïve estimator. In the main text, we estimate the (marginal) cross entropy with 2-gram correction in the following way. Breaking up the cross entropy as

$$H(p_{\mathbf{Y}}, q_{\mathbf{Y}}) = -\sum_{i=3}^{L-\ell} \mathbb{E}_p[\log q(Y_i|Y_{1:i-1})] + H(p_{Y_1 Y_2}, q_{Y_1 Y_2}). \tag{B.7}$$

For the first term, we use LLM generated $q(y_i|y_{1:i-1}, w_{BOS})$ as approximation. For the second term, we mitigate the bias from LLM estimation by combining it with Eq. (B.5) as $H(p_{Y_1 Y_2}, q_{Y_1 Y_2|w_{BOS}})/5 + 4\hat{H}_p^G(Y_1 Y_2)/5$. In Fig. B.3, we also present the result without this correction and show that this bias correction mostly affects the estimation at small lengths $L$, and does not alter the general scaling behavior. In addition, since the result from this bias-corrected direct estimator agrees with the vCLUB [40] estimator, we believe this correction is reasonable.

### B.V Additional Discussion on Mutual Information Estimation Methods

In the main text, we briefly discussed the limitations of traditional mutual information estimation methods for our high-dimensional, long-sequence setting. Here we provide additional technical details on why these methods are challenging to apply to our settings.

**Neural Estimators: MINE and InfoNCE.** Neural estimators like MINE [37] and InfoNCE [100] train deep neural networks as critics to estimate mutual information. Both methods can fundamentally be viewed as training unnormalized density estimators or density ratio estimators.

MINE uses the Donsker–Varadhan representation of KL divergence [104] and trains a critic $T_\theta(x, y)$ to maximize:

$$\mathbb{E}_{p(x,y)}[T_\theta(x, y)] - \log \mathbb{E}_{p(x)p(y)}[e^{T_\theta(x,y)}] \tag{B.8}$$

The optimal critic approximates the log density ratio $\log \frac{p(x,y)}{p(x)p(y)}$. However, this objective suffers from high variance and numerical instability when mutual information is large, which is especially challenging given the high-dimensional nature of long sequences we analyze.

InfoNCE uses noise-contrastive estimation with multiple negative samples:

$$I(X;Y) \geq \mathbb{E}\left[\log \frac{e^{f(x_i,y_i)}}{\frac{1}{K}\sum_{j=1}^{K} e^{f(x_i,y_j)}}\right] \tag{B.9}$$

where the expectation is over $K$ independent samples from the joint distribution. The bound is upper bounded by $\log K$, which means for our setting where mutual information can be on the order of thousands, this would require prohibitively large batch sizes to obtain accurate estimates.

Both methods require training critics from scratch to learn representations of natural language distributions, which could require datasets and computational resources comparable to training LLMs themselves.

Other variational bounds [36] face similar challenges.

**K-Nearest Neighbor Estimators.** K-nearest neighbor (K-NN) estimators [39] estimate mutual information based on distances between samples in joint and marginal spaces. While asymptotically unbiased and training-free, they also face challenges for text.

Text consists of discrete tokens that must be embedded into continuous spaces for K-NN estimation. Modern token embeddings have dimensions in the thousands, and for sequences of thousands of tokens, the combined dimensionality can make K-NN estimation impractical as the number of samples required for reliable K-NN estimates grows exponentially with dimension.

**Connection to Our LLM-Based Approach.** Our approach leverages pre-trained LLMs as density estimators, providing $q(y|x)$ directly through conditional probabilities and approximating $q(y)$ efficiently. This avoids training critics from scratch and the curse of dimensionality from distance-based estimation. We believe this is well-suited for our use case of analyzing long natural language sequences.

### B.VI  Estimation of Two-Point Mutual Information

In the main text, we included the results for two-point mutual information for completeness. In this section, we explain how the results are obtained.

Two-point mutual information estimation is more straightforward compared to bipartite mutual information, requiring only 1- and 2-gram statistics without LLM approximations. We estimate this quantity using entropy calculations for individual tokens and token pairs separated by distance $d$. Following [102], we employ their bias-reduced entropy estimator:

$$\hat{H}^G(X) = \hat{H}^G(Y) = \log N - \frac{1}{N}\sum_{m=1}^{M} n_m G(n_m), \tag{B.10}$$

where $N$ is the total number of tokens, $M$ is the vocabulary size, $n_m$ is the number of tokens whose token ID equals $m$, and $G(\cdot)$ is defined as

$$G(n) = \psi(n) + \frac{(-1)^n}{2}\left(\psi\left(\frac{n+1}{2}\right) - \psi\left(\frac{n}{2}\right)\right) \tag{B.11}$$

with $\psi(\cdot)$ the digamma function.

The entropy of pairs of tokens is estimated analogously, with the summation running over all ordered pairs of tokens $(m_i, m_j)$, resulting in the total number of terms quadratic in the vocabulary size. The mutual information is then estimated as

$$\hat{I}_d^{\text{TP}}(X;Y) = \hat{H}^G(X) + \hat{H}^G(Y) - \hat{H}^G(XY). \tag{B.12}$$

We note that this mutual information estimator exhibits systematic bias. The entropy estimator has a negative bias that increases (in absolute value) with dimension of the sample space $|\Omega|$. Since $|\Omega_X| = |\Omega_Y| = M$ where as $|\Omega_{XY}| = M^2$, the bias in $\hat{H}(XY)$ exceeds that in $\hat{H}(X) = \hat{H}(Y)$, leading to a positive bias in $\hat{I}_d$. This bias becomes particularly problematic at large distances $d$, where $H(XY) \approx H(X) + H(Y)$ and $I_d$ approaches zero. To mitigate this issue, we perform additional bias correction by fitting the systematic bias from the data (see Appx. B.VII for details).

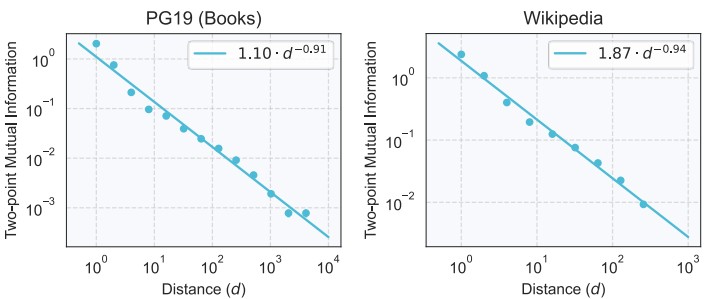

Figure B.4: Two-point mutual information scaling on PG19 and WIKIPEDIA datasets.

We apply this methodology to measure two-point mutual information on both the PG19 dataset and WIKIPEDIA, confirming power-law decay in both cases [Fig. B.4].

### B.VII    Bias Correction for Two-point Mutual Information

As discussed previously, the estimation of two-point mutual information can be calculated directly using the $n$-gram approximation [Eq. (B.5)], and compute the two-point mutual information as

$$\hat{I}_d^{\text{TP}}(X;Y) = \hat{H}^G(X) + \hat{H}^G(Y) - \hat{H}^G(XY). \tag{B.13}$$

As discussed in Appx. B.IV, this entropy estimator has a negative bias, whose magnitude depends on the ratio $|\Omega|/N$, with $|\Omega|$ the size of the corresponding sample space. Since the sample space for the joint distribution is larger, it has a larger negative bias, resulting in a positive bias in $\hat{I}$. When $d$ is small, this bias is relatively small compared to the mutual information itself. However, as $d$ becomes larger, $X$ and $Y$ become less correlated, and $H(XY) \to H(X) + H(Y)$. In this case, the estimator can be dominated by this bias, and fitting for the power-law exponent becomes impossible.

To mitigate this issue, we propose a bias-corrected estimator.

$$\hat{I}_d^{\text{TP,corrected}}(X;Y) = \hat{H}^G(X) + \hat{H}^G(Y) - \hat{H}^G(XY) - C, \tag{B.14}$$

where $C$ is an unknown positive constant that does not depend on the distance $d$, which accounts for the bias of the original estimator.

To obtain this bias correction term and fit the power-law exponent, we minimize the following loss function

$$\sum_d (\log{(\hat{I}_d^{\text{TP}} - C)} - (\log A - \alpha \log d))^2, \tag{B.15}$$

which is just $\hat{I}_d^{\text{TP}} = Ad^{-\alpha} + C$ fitted in log-log space. Then, we take the fitted $C$ as the systematic bias and $\alpha$ as the fitted power-law exponent.

In Fig. B.5, we compare the corrected and uncorrected two-point mutual information as a function of $d$ (only the corrected version is shown in the main text). Without the bias correction, the data appear to have larger long-range dependencies, but after the bias correction, all points lie on a straight line in a log-log plot. The bias correction constant is much smaller than the entropies involved in the calculation, even the smallest two-token entropy measured is 12.5, at least two orders of magnitude larger than the fitted bias correction. In addition, the fact that a single variable added to the fitting function can fit the data so well suggests the bias correction is reasonable and highly effective.

We note that on WIKIPEDIA, we were only able to measure the two-point mutual information up to $d = 256$, due to limited long-context length data in WIKIPEDIA.

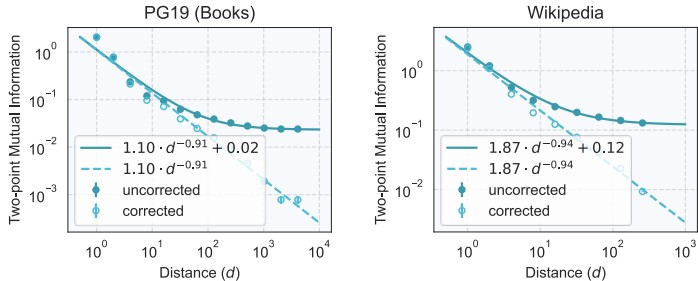

Figure B.5: Effect of bias correction for two-point mutual information. The bias causes a plateau at large distances.

## B.VIII    Failures of Two-point Mutual Information

In this section, we further demonstrate the relation between two-point and bipartite mutual information, and why two-point mutual information cannot properly capture the full multi-token dependencies needed for modeling natural language.

In existing literature, the scaling of two-point mutual information has been used to demonstrate the existence of long-range dependencies in natural language. In particular, it is believed that short-range dependencies are characterized by an exponential decay in two-point mutual information, as seen typically in finite-state Markov chains, and the existence of power-law decay in two-point mutual information in natural language indicates non-trivial long-range dependencies. Although this perspective is correct if one defines the existence of long-range dependence as as the existence of non-exponential-decay two-point mutual information, this definition does not properly account for the mutual information between token pairs when other tokens are present. This can be made more clear by considering the following decomposition of bipartite mutual information.

For a sequence of tokens $W_{1:L}$ with $X_{1:\ell} = W_{1:\ell}$ and $Y_{1:L-\ell} = W_{\ell+1:L}$, the bipartite mutual information reads

$$I^{\mathrm{BP}}_{\ell;L} = I(X_{1:\ell}; Y_{1:L-\ell}). \tag{B.16}$$

Standard information theory allows mutual information to be decomposed as

$$I(XZ;Y) = I(X;Y) + I(Z;Y|X), \tag{B.17}$$

where $I(Z;Y|X)$ is the conditional mutual information between $Z$ and $Y$ given $X$. Using this relation repeatedly, the bipartite mutual information can be decomposed as

$$
\begin{aligned}
I^{\mathrm{BP}}_{\ell;L} &= I(X_{1:\ell}; Y_{1:L-\ell}) \\
&= I(X_1;Y_1) + I(X_2;Y_1|X_1) + I(X_1;Y_2|Y_1) + I(X_2;Y_2|X_1Y_1) + \cdots \\
&= \sum_{i=1}^{\ell}\sum_{j=1}^{L-\ell} I(X_i;Y_j|X_{1:i-1}Y_{1:j-1}) \\
&\neq \sum_{i=1}^{\ell}\sum_{j=1}^{L-\ell} I(X_i;Y_j) = \sum_{i=1}^{\ell}\sum_{j=1}^{L-\ell} I^{\mathrm{TP}}_{j-i+\ell}.
\end{aligned}
\tag{B.18}
$$

In fact, it is in general not even clear whether the conditional mutual information $I(X_i;Y_j|X_{1:i-1}Y_{1:j-1})$ is greater or less than the marginal mutual information $I(X_i;Y_j)$. Therefore, as demonstrated here, when considering dependencies between blocks of text, a simple aggregation of the two-point mutual information gives a very incomplete picture. Due to this reason, weakly correlated systems, such as the example mentioned in Sec. 4.4 could exhibit seeming strong long-range two-point dependencies, and systems with very different bipartite mutual information, could share very similar two-point information scaling, as we will show later in Appx. C.

# C   Multivariate Gaussian Distributions

In the main text, we considered two families of multivariate Gaussian distributions of different sequence lengths to demonstrate the distinction between bipartite and two-point mutual information scalings. In particular, one is designed to mimic natural language, both in terms of the sub-volume law growth of the bipartite mutual information and the power-law decay of two-point mutual information. This family of distributions is also used to empirically verify our theory on $L^2M$ condition for different LLM architectures. The other is designed to have the same two-point mutual information scaling, but very different bipartite mutual information scaling, showcasing that one can have distributions with the same two-point mutual information scaling, but drastically different bipartite mutual information scalings.

## C.I   Mutual Information Scalings

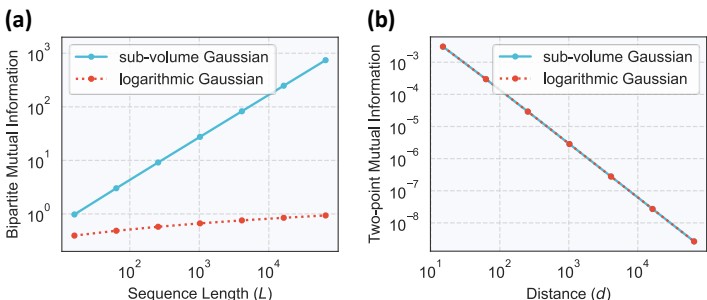

Figure C.6: Bipartite and two-point mutual information of the two families of Gaussian distributions.

Before showing the construction details, we first present both the bipartite and two-point mutual information scalings of the two families of Gaussian distributions. As shown in Fig. C.6, they have drastically different bipartite mutual information scaling—one has a power-law relation and the other logarithmic—but their antipodal two-point mutual information scaling is identical. This further demonstrates that two-point mutual information alone gives incomplete information of the multi-token long-range dependencies present in sequence data, and the simple aggregation of two-point mutual information does not tell the full picture of the bipartite mutual information. In fact, one can construct distributions with the same power-law decay in two-point mutual information, but has a constant bipartite mutual information scaling as well.

## C.II   Construction

Let's start by considering the family of distributions with sub-volume law growth. The distributions are constructed in a hierarchical manner.

We start at the first layer, with four independent standard Gaussian random variables $(X_1, X_2, X_3, X_4)$. Then, define the change-of-coordinate matrix

$$\mathcal{M} = \begin{pmatrix} \gamma & \gamma & \gamma & \rho \\ -\gamma & \gamma & -\gamma & \rho \\ -\gamma & -\gamma & \gamma & \rho \\ \gamma & -\gamma & -\gamma & \rho \end{pmatrix}, \tag{C.19}$$

where we choose $\gamma = \sqrt{5}/4$ and $\rho = 1/4$. The output of the first layer is defined as

$$\boldsymbol{Y} = \mathcal{M}\boldsymbol{X}, \tag{C.20}$$

where the random variables are now correlated. It is easy to verify that this operation only changes the off-diagonal elements in the covariance matrix, and leaves the diagonal elements unaffected.

For the second layer and up, we first stack three independently sampled copies from the previous layer and attach additional independent standard Gaussian random variables as the fourth elements as

$$\mathcal{X} = \begin{pmatrix} Y_{1,1} & Y_{1,2} & Y_{1,3} & W_1 \\ Y_{2,1} & Y_{2,2} & Y_{2,3} & W_2 \\ Y_{3,1} & Y_{3,2} & Y_{3,3} & W_3 \\ \vdots & \vdots & \vdots & \vdots \end{pmatrix}, \tag{C.21}$$

where $Y_{i,j}$ refers to the $i$th output from previous layer of the $j$th copy, and $W_i$ refers to the $i$th newly sampled standard Gaussian random variable.

Note that at this point, all rows are independent from each other; therefore we apply the change of coordinate matrix $\mathcal{M}$ at each row to correlate them. The matrix is then flattened to obtain $\boldsymbol{Z}$

$$\boldsymbol{Z} = (Z_{1,1}, Z_{1,2}, Z_{1,3}, Z_{1,4}, Z_{2,1}, Z_{2,2}, Z_{2,3}, Z_{2,4}, Z_{3,1}, Z_{3,2}, Z_{3,3}, Z_{3,4}, \cdots), \tag{C.22}$$

where the subscripts denote the variables' original position in the matrix. Before outputting from this layer, we perform an addition operation to each pair of random variables $(Z_{i,4}, Z_{i+1,1})$, by applying a coordinate transformation that modifies their correlations as

$$\operatorname{corr}(Z_{i,4}, Z_{i+1,1}) \rightarrow \frac{2}{5}(\operatorname{corr}(Z_{i,3}, Z_{i,4}) + \operatorname{corr}(Z_{i+1,1}, Z_{i+1,2})) + \frac{1}{5}. \tag{C.23}$$

This operation may seem arbitrary, but it is crucial to introduce correlations that give a linear ordering of the random variables. Without this operation, the distribution simply forms a tree structure.

Now, we can truncate the construction at different layers $l$ and form a family of distributions with different sequence lengths $L = 4^l$. In Fig. C.6 of the main text, we consider up to 8 layers, and in Fig. 3 and 4, we consider $l = 4$, 5 and, 6.

The second family of distributions is constructed analogously. The only difference is that we replace Eq. (C.21) with a single copy of $\boldsymbol{Y}$ and three independent copies of $\boldsymbol{W}$ as

$$\mathcal{X} = \begin{pmatrix} Y_1 & W_{1,1} & W_{1,2} & W_{1,3} \\ Y_2 & W_{2,1} & W_{2,2} & W_{2,3} \\ Y_3 & W_{3,1} & W_{3,2} & W_{3,3} \\ \vdots & \vdots & \vdots & \vdots \end{pmatrix}, \tag{C.24}$$

### C.III  Properties

These two series of distributions have many nice properties, in addition to their bipartite and two-point mutual information scalings. In addition, these constructions directly defines the multi-variate probability distribution due to their Gaussian nature. This allows for exact calculations of conditional probability distributions for training LLMs, as well as direct computation of the bipartite and two-point mutual information without LLM approximations.

## D  Model State for Storing Past Information

In Definition 5.1 in the main text, we give a concrete definition of "model state for storing past information" as history state, and claim that it is the past key-value pairs for transformers and recurrent state for SSMs and RNNs. Here, we explain them in more detail.

### D.I  Transformers

In transformers, only the attention block mixes information among different tokens, therefore we only need to analyze the behavior of the attention block. We will be assuming the existence of the causal mask, as our theory mainly applied to autoregressive LLMs. Denoting the input and output of the attention layer as $\boldsymbol{x}$ and $\boldsymbol{y}$ (notice they are no longer two parts of a sequence), the self-attention mechanism is defined as

$$\boldsymbol{y} = \operatorname{softmax}((W_q \boldsymbol{x})(W_k \boldsymbol{x})^T) W_v \boldsymbol{x}, \tag{D.25}$$

where $W_q$, $W_k$ and $W_v$ are the weight matrices. For simplicity, we drop the usual $\sqrt{h_{\mathrm{dim}}}$ normalization and the output weight matrix, as they are irrelevant to our discussion.

Separating the calculation for each token, the mechanism can be rewritten as

$$y_i = \frac{\sum_{j=1}^{i} e^{(W_q x_i)(W_k x_j)} W_v x_j}{\sum_{j'=1}^{i} e^{(W_q x_i)(W_k x_{j'})}} = \frac{e^{(W_q x_i)(W_k x_i)} W_v x_i + \sum_{j=1}^{i-1} e^{(W_q x_i) k_j} v_j}{e^{(W_q x_i)(W_k x_i)} + \sum_{j'=1}^{i-1} e^{(W_q x_i) k_{j'}}}, \tag{D.26}$$

where $k_j = W_k x_j$ and $v_j = W_v x_j$ are keys and values which we sum over past tokens. Clearly, the attention output only depends on the current token $x_i$ and the past key-value pairs $k_{1:i-1}$ and $v_{1:i-1}$. This arguments extends to all $y_k$ with $k \geq i$, where all $y_k$'s dependency on $x_{1:i-1}$ is via $k_{1:i-1}$ and $v_{1:i-1}$. Therefore, key-value pairs form the history state, and their size grows linearly with input sequence length. We note that Eq. (D.26) also describes how key-value caching works.

## D.II State Space Models and RNNs

State space models (SSMs) and RNNs, on the other hand, are easier to analyze. These models in general all have some recurrent state with a fixed size, and some mechanism to update the state when a new token is observed. The output depends only on the previous recurrent state, and the current token. They can in general be written in the following way.

$$\begin{aligned} h_i &= f(h_{i-1}, x_i), \\ y_i &= g(h_{i-1}, x_i), \end{aligned} \tag{D.27}$$

for some update function $f$ and output function $g$. It is obvious that the history state is exactly this recurrent state (or the collection of recurrent states of different layers), which does not grow with the input sequence.

We note that this discussion also applies to linear attention models, whose key-value pairs can be merged into a recurrent state with fixed size, due to the replacement of softmax function. Test time training (TTT) models can also be included in this discussion. They can be viewed as RNNs with inner model parameters as recurrent state, and test time training process as update function.

## D.III Other Architectures

For other models, such as sparse transformers or some compression-based models, the analysis has to be performed separately. Nevertheless, the L$^2$M framework is general: after identifying the history state, one can always compare its scaling with the bipartite mutual information scaling to see whether the model is capable of capturing the long-range dependencies in the data.

# E Proofs of Theorem 5.2

We provide three proofs of Theorem 5.2 under different assumptions, demonstrating the universality and robustness of the result. Importantly, all three sets of assumptions are extremely mild and directly reflect realistic conditions in modern neural networks, whether through the discrete nature of floating-point arithmetic, empirically observed geometric properties of neural representations, or basic continuity requirements.

## E.I Proof Under Discreteness Assumption

The discreteness assumption is already quite reasonable in practice. Modern neural networks use floating-point representations, which are inherently discrete with finite precision. Moreover, neural networks have been shown to retain strong performance even under aggressive quantization, demonstrating that discrete representations with limited precision are sufficient to capture the essential information. This discreteness assumption thus provides a natural and practical starting point for our proof.

**Theorem E.1** (Theorem 5.2, Discrete Version). *Assume the history state $z_\ell$ takes discrete values. Then a model's capacity to capture bipartite mutual information is bounded by the size of its history state as*

$$I_{\ell;L}^{\mathrm{BP},q} \leq C \cdot \dim(z_\ell) + \log(M) \tag{E.28}$$

*where $C$ is some constant and $M$ denotes the vocabulary size.*

*Proof.* By the data processing inequality: $I^q(X_{1:\ell}; Y_{1:L-\ell}) \leq I^q(\boldsymbol{Z}_\ell X_\ell; Y_{1:L-\ell})$. This is upper bounded by the entropy: $H^q(\boldsymbol{Z}_\ell X_\ell)$, which is further upper bounded by $H^q(\boldsymbol{Z}_\ell) + H^q(X_\ell) \leq C \cdot \dim(\boldsymbol{z}_\ell) + \log(M)$, where the last inequality follows from the bound on entropies of discrete variables. $\square$

### E.II Proof Under Almost Orthogonal Directions (AOD) Assumption

The discreteness assumption can be relaxed if we instead assume the following observed fact about neural networks: neural networks store distinct information in almost orthogonal directions (AODs) of the hidden state [105–107].

**Theorem E.2** (Theorem 5.2, AOD Version). *Assume neural networks store distinct information in almost orthogonal directions (AODs) of the hidden state. Then a model's capacity to capture bipartite mutual information is bounded as*

$$I_{\ell;L}^{\mathrm{BP},q} \leq C \cdot \dim(\boldsymbol{z}_\ell) + \log(M) \tag{E.29}$$

*where $C$ is some constant and $M$ denotes the vocabulary size.*

*Proof.* An autoregressive neural network's dependency on past tokens is through the intermediate variable $\boldsymbol{z}_\ell = \boldsymbol{f}(x_{1:\ell-1})$ such that $q(\boldsymbol{y}|\boldsymbol{x}) := q(\boldsymbol{y}|x_\ell, \boldsymbol{z}_\ell)$. This can be viewed as the process $\boldsymbol{X} \to (\boldsymbol{Z}_\ell, X_\ell) \to \boldsymbol{Y}$. According to the data processing inequality,

$$I^q(X_{1:\ell}; Y_{1:L-\ell}) \leq I^q(\boldsymbol{Z}_\ell, X_\ell; Y_{1:L-\ell}) \leq \mathcal{H}^q(\boldsymbol{Z}_\ell, X_\ell) \leq \mathcal{H}^q(\boldsymbol{Z}_\ell) + H^q(X_\ell), \tag{E.30}$$

where we use $\mathcal{H}$ to denote a generalized notion of entropy which measures the amount of information that can be stored in $\boldsymbol{Z}_\ell$. We only care about the scaling of $\mathcal{H}$, so its exact definition is irrelevant to our discussion.

Under the AOD assumption, neural networks store distinct information in almost orthogonal directions of the hidden state. Therefore, $\mathcal{H}$ should scale at most logarithmically with respect to the number of AODs as the state size increases. According to the Kabatjanskii–Levenstein bound [108, 109], given an error tolerance $\varepsilon$, the number of AODs is upper bounded by $\exp(f(\varepsilon) \cdot \dim(\boldsymbol{z}_\ell))$ for some function $f$ that depends purely on the error threshold. Therefore, the generalized entropy scales as $\mathcal{H}^q(\boldsymbol{Z}_\ell) \precsim \log \exp(f(\varepsilon) \cdot \dim(\boldsymbol{z}_\ell)) \sim \dim(\boldsymbol{z}_\ell)$. Since $H^q(X_\ell) \leq \log(M)$ where $M$ is the vocabulary size, we conclude

$$I_{\ell;L}^{\mathrm{BP},q} \leq C \cdot \dim(\boldsymbol{z}_\ell) + \log(M). \tag{E.31}$$

$\square$

### E.III Proof Under Lipschitz Continuity Assumption

The theorem can also be proved assuming only certain Lipschitz continuity conditions on the neural network.

**Theorem E.3** (Theorem 5.2, Lipschitz Version). *Assume the history state mapping $\boldsymbol{f} : x_{1:\ell-1} \mapsto \boldsymbol{z}_\ell$ satisfies $\left\| \boldsymbol{f}(x_{1:\ell-1}) - \boldsymbol{f}(x'_{1:\ell-1}) \right\|_2 \leq K_f \mathbb{1}(x_{1:\ell-1} \neq x'_{1:\ell-1})$ and the neural network is entropy-Lipschitz, satisfying $|H^q(\boldsymbol{Y}|\boldsymbol{z}_\ell) - H^q(\boldsymbol{Y}|\boldsymbol{z}'_\ell)| \leq K_H \|\boldsymbol{z}_\ell - \boldsymbol{z}'_\ell\|_2$. Then a model's capacity to capture bipartite mutual information is bounded as*

$$I_{\ell;L}^{\mathrm{BP},q} \leq C \cdot \dim(\boldsymbol{z}_\ell) + \log(M) \tag{E.32}$$

*where $C$ is some constant and $M$ denotes the vocabulary size.*

*Proof.* We start with the data processing inequality and rewrite the bound as

$$\begin{aligned} I^q(X_{1:\ell}; Y_{1:L-\ell}) &\leq I^q(\boldsymbol{Z}_\ell, X_\ell; Y_{1:L-\ell}) \\ &= I^q(\boldsymbol{Z}_\ell; Y_{1:L-\ell}) + I^q(X_\ell; Y_{1:L-\ell}|\boldsymbol{Z}_\ell) \\ &\leq I^q(\boldsymbol{Z}_\ell; Y_{1:L-\ell}) + \log(M), \end{aligned} \tag{E.33}$$

where the last inequality uses $I^q(X_\ell; Y_{1:L-\ell}|\boldsymbol{Z}_\ell) \leq H^q(X_\ell) \leq \log(M)$, with $M$ being the vocabulary size.

The history state is a function of the input tokens $\boldsymbol{z}_\ell = \boldsymbol{f}(x_{1:\ell-1})$, with $x_{1:\ell-1} \in \{1, 2, \ldots, M\}^{\ell-1}$. Under our assumption on $\boldsymbol{f}$, $\boldsymbol{z}_\ell$ lives in a $d$-dimensional ball of radius $K_f$, where $d = \dim(\boldsymbol{z}_\ell)$.

Consider a quantization $\boldsymbol{Q}(\boldsymbol{z}_\ell)$ that maps each $\boldsymbol{z}_\ell$ to the nearest point in an $\varepsilon$-covering of this ball. Then

$$I^q(\boldsymbol{Z}_\ell; \boldsymbol{Y}) = I^q(\boldsymbol{Q}(\boldsymbol{Z}_\ell); \boldsymbol{Y}) + H^q(\boldsymbol{Y}|\boldsymbol{Q}(\boldsymbol{Z}_\ell)) - H^q(\boldsymbol{Y}|\boldsymbol{Z}_\ell). \tag{E.34}$$

By the entropy-Lipschitz assumption, $H^q(\boldsymbol{Y}|\boldsymbol{Q}(\boldsymbol{Z}_\ell)) - H^q(\boldsymbol{Y}|\boldsymbol{Z}_\ell) \leq K_H\varepsilon$. Since $\boldsymbol{Q}(\boldsymbol{Z}_\ell)$ is discrete and takes at most $(2K_f/\varepsilon)^d$ values (by a covering number argument), we have $I^q(\boldsymbol{Q}(\boldsymbol{Z}_\ell); \boldsymbol{Y}) \leq H^q(\boldsymbol{Q}(\boldsymbol{Z}_\ell)) \leq d\log(2K_f/\varepsilon)$.

Therefore, $I^q(\boldsymbol{Z}_\ell; \boldsymbol{Y}) \leq d\log(2K_f/\varepsilon) + K_H\varepsilon \leq C \cdot d$ for some constant $C$ (by choosing $\varepsilon$ appropriately). This concludes

$$I_{\ell;L}^{\mathrm{BP},q} \leq C \cdot d + \log(M) = C \cdot \dim(\boldsymbol{z}_\ell) + \log(M). \tag{E.35}$$

$\square$

### E.IV Discussion

We believe this theorem is more universal and can be proved in additional ways, such as by connecting it to channel capacity and potentially showing $I_{\ell;L}^{\mathrm{BP},q} \leq d\log(1 + \mathrm{SNR}) + \log(M)$. We also believe the theorem can be established with more relaxed assumptions, similar to how information dimension is proved to be the upper bound of lossless compression of continuous random variables [110]. However, additional proofs are beyond the scope of this work, and the three proofs provided should already be applicable in any practical settings.

## F Additional Details on Experimental Setup

In this section, we provide detailed information about our experimental setup, including dataset construction, model configurations, training procedures, and evaluation metrics.

### F.I Synthetic Gaussian Distribution Dataset

For experiments on the multivariate Gaussian distribution, we use the sub-volume Gaussian distributions described in Appx. C, which exhibits power-law bipartite mutual information scaling with an exponent of $0.79$. To fully stress the LLMs, we stack 64 copies of the distribution and group the 64 Gaussian variables at each position to form a single token. More specifically, an example sample looks like

$$\begin{aligned}\boldsymbol{W} &= (W_1, W_2, \ldots, W_L) \\ &:= ((Z_{1,1}, Z_{1,2}, \ldots, Z_{1,64}), (Z_{2,1}, Z_{2,2}, \ldots, Z_{2,64}), \ldots, (Z_{L,1}, Z_{L,2}, \ldots, Z_{L,64})),\end{aligned} \tag{F.36}$$

where the two subscripts $(i, j)$ refer to the $i$th random variable from the $j$th copy. In this way, the bipartite mutual information matches better with natural language, not only in scaling, but also in magnitude (multiplicative constant). We additionally prepend an all-zero token $W_0$ to each sample to mimic the effect of the BOS token.

In order to process continuous random variables, we replace the embedding layers of GPT2 and Mamba(2) models with two-layer MLPs. For output, since all the conditional distributions are also Gaussian, we use a different two-layer MLP to output the 64 conditional means $\mu_{q_{Z_{i,j}|Z_{0:i-1,j}}}$ and standard deviations $\sigma_{q_{Z_{i,j}|Z_{0:i-1,j}}}$.

As discussed in Appx. C.III, due to the analytical construction, the Gaussian distribution permits efficient calculation of conditional probabilities. Therefore, instead of simply training the neural networks with negative log likelihood on samples alone, we use the average conditional KL-divergence estimated as

$$D_{KL}(p\|q_\theta) =$$

$$\mathbb{E}_{p_Z}\left[\frac{1}{L}\sum_{i=1}^{L}\frac{1}{64}\sum_{j=1}^{64}\left(\log\frac{\sigma_{q_{Z_{i,j}|Z_{0:i-1,j}}}}{\sigma_{p_{Z_{i,j}|Z_{0:i-1,j}}}} + \frac{\sigma_{p_{Z_{i,j}|Z_{0:i-1,j}}}^2 + (\mu_{q_{Z_{i,j}|Z_{0:i-1,j}}} - \mu_{p_{Z_{i,j}|Z_{0:i-1,j}}})^2}{2\sigma_{q_{Z_{i,j}|Z_{0:i-1,j}}}^2} - \frac{1}{2}\right)\right] \tag{F.37}$$

to reduce sampling variance.

## F.II    Natural Language Dataset (PG19)

For the PG19 dataset, we train on standard average negative log likelihood. We first split the dataset into samples with a length of approximately 1.2 times the target length, ensuring each sample starts at the beginning of a sentence. We then train the models for 5 epochs (approximately 450k iterations) with a batch size of 16,384 tokens. To maintain consistency across different models, we always use the same tokenizer from GPT-Neo-X [111].

## F.III    Training Configuration

For the Gaussian distribution training, during each iteration, we use a batch size of 4 (4 times sequence length number of tokens) with freshly generated samples, meaning we never reuse any sample. We therefore have effectively a single epoch, thanks to the "infinite" dataset size. We train all neural networks using the AdamW optimizer and a cosine decay scheduler with warmup. We use a peak learning rate of $5 \times 10^{-5}$, a weight decay of 0.01, 2000 warmup steps, and 500,000 training steps in total. The results reported are at the end of training.

For the PG19 dataset experiments, we use similar hyperparameters: AdamW optimizer with a cosine decay scheduler with warmup, peak learning rate of $5 \times 10^{-5}$, weight decay of 0.01, 2000 warmup steps, and 500,000 steps in total. The results reported are at the end of training using a separate evaluation dataset containing 10,000 samples.

## F.IV    Evaluation Metrics

In this paper, we report results on the position-wise conditional KL-divergence

$$D_{KL,i} = D_{KL}(p_{W_i|W_{1:i-1}} || q_{W_i|W_{1:i-1}}) = \mathbb{E}_p \left[ \log p(W_i|W_{1:i-1}) - \log q(W_i|W_{1:i-1}) \right], \quad \text{(F.38)}$$

average KL-divergence

$$D_{KL}^{\text{avg}} = \frac{1}{L} \sum_{i=1}^{L} D_{KL,i}, \quad \text{(F.39)}$$

and position-wise conditional NLL

$$\text{NLL}_i = -\mathbb{E}_p \left[ \log q(W_i|W_{1:i-1}) \right]. \quad \text{(F.40)}$$

One can also define an average NLL as

$$\text{NLL}^{\text{avg}} = \frac{1}{L} \sum_{i=1}^{L} \text{NLL}_i, \quad \text{(F.41)}$$

which we use in Appx. G.

### F.IV.1    Understanding the Behavior of KL-divergence and NLL

It is important to understand how KL-divergence and NLL behave differently as token position increases. Using the relationship between cross-entropy and KL-divergence from Eq. (5), we can decompose the conditional NLL as

$$\text{NLL}_i = D_{KL,i} + H^p(W_i|W_{1:i-1}), \quad \text{(F.42)}$$

where $H^p(W_i|W_{1:i-1}) = -\mathbb{E}_p[\log p(W_i|W_{1:i-1})]$ is the conditional entropy of the true distribution at position $i$.

This decomposition reveals why KL-divergence and NLL exhibit opposite trends with token position. As token position increases, the conditional entropy $H^p(W_i|W_{1:i-1})$ typically decreases because more context is available, making the next token more predictable. In natural language, this reflects that with more preceding text, there is less uncertainty about what comes next. Meanwhile, the conditional KL-divergence $D_{KL,i}$ often increases with position, because learning all long-range dependencies becomes more challenging, resulting in worse model estimations compared to the true conditional distribution.

For models with sufficient capacity relative to sequence length, the decrease in conditional entropy dominates, causing NLL to decrease with position despite increasing KL-divergence. However, for models with insufficient capacity (such as fixed-state models at long sequence lengths), the KL-divergence can increase rapidly enough that NLL plateaus or even increases at later positions. This behavior is precisely what we observe in our experiments with Mamba models in Fig. 4, where Mamba's NLL plateaus at later positions while transformers' NLL continues to improve.

The same reasoning applies to average quantities: $\text{NLL}^{\text{avg}}$ typically decreases with sequence length $L$ as the average conditional entropy decreases, while $D_{KL}^{\text{avg}}$ may increase if the model's capacity becomes insufficient relative to the growing sequence length.

When reporting conditional NLL, we smooth the curves using a small window around nearby tokens to reduce noise in the results.

### F.V Model Configurations

In Tables F.1 and F.2, we include the model configurations and sequence lengths for all experiments performed in this paper.

Table F.1: Models and configurations for synthetic dataset experiments.

| Model | num_hidden_layers | hidden_size | seq_len |
| --- | --- | --- | --- |
| GPT2 | 12 | 768 | 256,1024,4096 |
| GPT2-medium | 24 | 1024 | 256,1024,4096 |
| GPT2-large | 36 | 1280 | 256,1024,4096 |
| Mamba-50m | 12 | 512 | 64,256,1024,4096,16384 |
| Mamba-70m | 24 | 512 | 64,256,1024,4096,16384 |
| Mamba-130m | 24 | 768 | 64,256,1024,4096,16384 |
| Mamba-370m | 48 | 1024 | 64,256,1024,4096 |
| Mamba-790m | 48 | 1536 | 64,256,1024,4096 |
| Mamba-1.4b | 48 | 2048 | 64,256,1024,4096 |
| Mamba2-130m | 24 | 768 | 256,1024,4096 |
| Mamba2-370m | 48 | 1024 | 256,1024,4096 |
| Mamba2-790m | 48 | 1536 | 256,1024,4096 |

Table F.2: Models and configurations for PG19 experiments.

| Model | num_hidden_layers | hidden_size | seq_len |
| --- | --- | --- | --- |
| GPT2 | 12 | 768 | 4096,8192 |
| GPT2-medium | 24 | 1024 | 4096,8192 |
| GPT2-large | 36 | 1280 | 4096,8192 |
| Mamba-130m | 24 | 768 | 4096,8192 |
| Mamba-370m | 48 | 1024 | 4096,8192 |
| Mamba-790m | 48 | 1536 | 4096,8192 |

### F.VI Computational Resources and Implementation Details

Our experiments are performed primarily on H100 GPUs, with varying VRAM sizes between 80GB and 96GB. Some experiments use A100 GPUs with 80GB VRAM instead. We use the `vLLM` library [68] when running inference to estimate the mutual information scaling. For both LLaMA 3.1 405B and DeepSeek V3, we run the FP8 version using 8 H100 GPUs (with 96GB VRAM each). The model weights and configurations are downloaded from HuggingFace [112].

When training GPT and Mamba(2) models on the Gaussian distribution, we use our custom library developed in `PyTorch` [113]. When training GPT and Mamba models on the PG19 dataset, we use the trainer from the HuggingFace `transformers` library. All models are initialized from scratch, with model configurations taken from HuggingFace. All training experiments are performed on individual H100 and A100 GPUs with FP32 precision to avoid possible training failures. Although

training with FP16 would make the experiments run faster, it should not affect the actual results. We note that for Mamba2, we use the official implementation instead of the HuggingFace version. For the GPT2 experiments on PG19, we re-implement the attention mechanism with FlexAttention [114] to save memory, as the official FlashAttention [64–66] does not support FP32 precision.

## F.VII  Code Availability

The code for reproducing our mutual information estimation and the PG19 results is available at `https://github.com/LSquaredM/mutual_info_scaling_law`.

## G  Additional Experimental Results

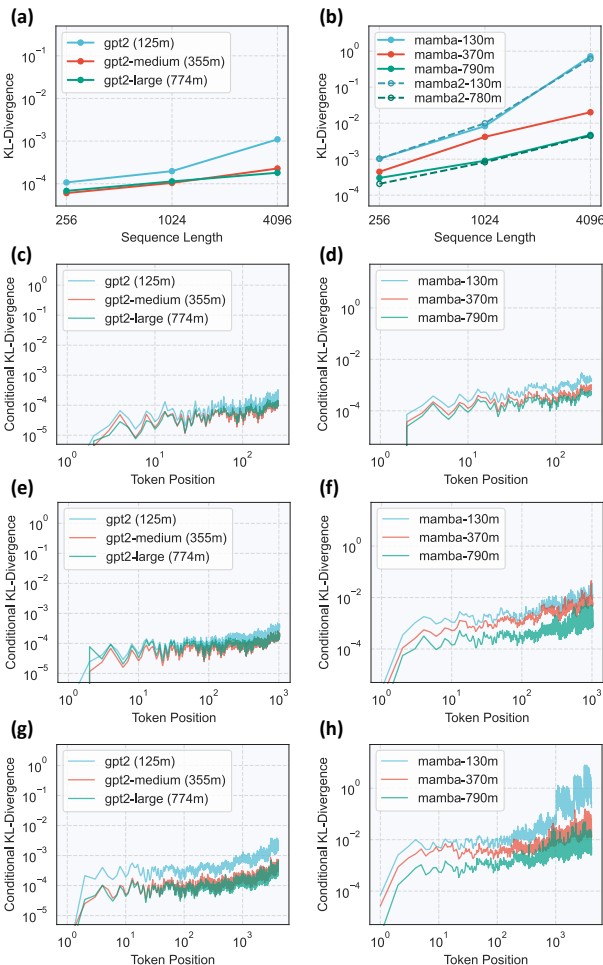

Figure G.7: Evaluation of KL-divergence across model architectures trained on sub-volume Gaussian distributions. (a, b) Average KL-divergence per token for models trained on different sequence lengths [same as Fig. 3 (a, b)]. (c, d) Position-wise conditional KL-divergence for models trained on sequence length 256. (e, f) Position-wise conditional KL-divergence for models trained on sequence length 1024. (g, h) Position-wise conditional KL-divergence for models trained on sequence length 4096 [same as Fig. 3 (c, d)]. Lower values indicate better performance.

In this section, we show additional experimental results. In Fig. G.7, we include positional-wise conditional KL-divergences of models trained on sub-volume Gaussian distributions with sequence length 256 (c, d), 1024 (e, f), and 4096 (g, h). As clearly demonstrated in the figure, for short sequence lengths, Mamba maintains similar performances to GPT2; Mamba models of different sizes also appear to have a smaller performance gap. However, as we go to longer sequence lengths,

smaller Mamba models starts to fail, while GPT2 always maintain relatively stable performances, consistent with our theory.

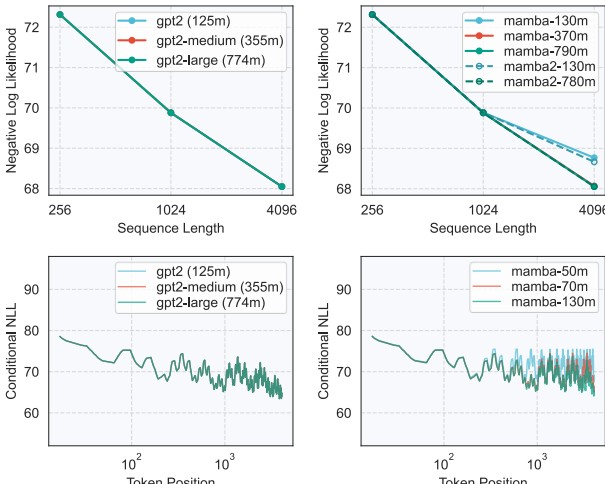

Figure G.8: Negative log likelihood (NLL) across model architectures trained on sub-volume Gaussian distributions (a, b) Average NLL per token for models trained on different sequence lengths. (c, d) Position-wise conditional NLL for models trained on sequence length 4096. Lower values indicate better performance.

In Fig. G.8, we show the negative log likelihood (NLL) of models trained on sub-volume Gaussian distributions. We note that, because NLL combines the KL-divergence with the intrinsic entropy of the underlying distribution (the average and position-wise conditional of which decays as sequence lengths), the differences between model performances are less visible. It's worth noting that, since Gaussian random variables are continuous, NLL values can differ by an arbitrary additive constant by rescaling the distribution. Therefore, the exact values of conditional NLL do not carry intrinsic meaning, though relative comparisons (which is exactly the same as the KL-divergence) between models remain valid.

In Fig. G.9, we show the position-wise conditional negative log likelihood (NLL) of models trained on the PG19 dataset [90] with 4096-token sequences. The results here is consistent with the 8192-token-sequence results in the main text.

# H   Limitations

Our theoretical framework specifically examines models' capacity to capture long-range dependencies through the lens of bipartite mutual information and does not address other aspects of language modeling, such as reasoning capabilities or world knowledge. The $L^2M$ condition establishes necessary but not sufficient conditions for effective long-context modeling. Understanding how this theoretical capacity translates to actual downstream task performance remains an important open question. The relationship likely depends on additional factors including optimization dynamics, architectural inductive biases, and task-specific requirements. Systematic evaluation across diverse long-context benchmarks represents a crucial next step to clarify these relationships and identify any gaps between theoretical capability and practical performance.

While our empirical validation on synthetic Gaussian distributions with controlled mutual information scaling provides clean verification of the theoretical predictions, it may not capture all complexities present in natural language. Our theory focuses on autoregressive language models, which remains broadly applicable as even diffusion-based approaches typically employ autoregressive generation for extended sequences in practice. Nonetheless, exploring whether similar information-theoretic principles govern fundamentally different generative paradigms or multimodal models represents an interesting direction for future work.

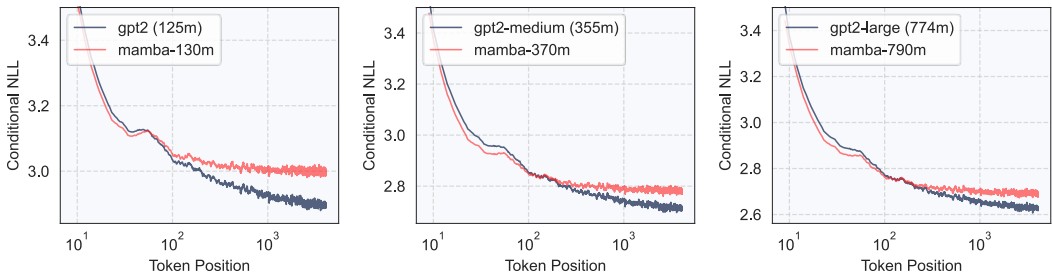

Figure G.9: Position-wise conditional negative log likelihood (NLL) evaluation for models trained on 4096-token sequences on the PG19 dataset [90].

The methodology we employ, using LLMs as density estimators for mutual information measurement, represents a practical approach given the severe challenges of high-dimensional estimation in long sequences. Alternative methods like K-NN and neural estimators face fundamental difficulties with dimensionality and sequence length. While our approach yields consistent power-law behavior across different models and estimators, both methods likely underestimate the true exponent, and developing more accurate estimation techniques remains an important challenge.

Our evaluations rely primarily on open-source models; further verification using state-of-the-art closed-source models would provide additional validation.

## I  Broader Impact

This work advances our theoretical understanding of how language models process long-range dependencies, with implications for the design and deployment of more efficient LLM architectures. By establishing the $L^2M$ condition, we provide a principled framework for evaluating an architecture's fundamental capacity for long-context modeling. This could lead to more efficient models that maintain effectiveness while reducing computational resources, potentially decreasing the environmental footprint of training and inference. Our findings may influence the development of specialized architectures for tasks requiring long-context understanding, such as legal document analysis, scientific research, and complex reasoning.

However, improved long-context modeling could also amplify existing challenges in LLMs, including the propagation of bias over longer contexts and enhanced capabilities for generating persuasive misinformation. Research applying the $L^2M$ framework should consider these ethical dimensions, particularly how improvements in long-range dependency modeling might affect model safety, fairness, and the verifiability of model outputs.

