# OpenReview forum: "L$^2$M: Mutual Information Scaling Law for Long-Context Language Modeling"
_NeurIPS.cc/2025/Conference — NeurIPS 2025 poster_

### Official Review · Reviewer_gdYy · 2025-06-29

**Clarity:** 3
**Significance:** 3
**Originality:** 3
**Rating:** 5
**Confidence:** 3

**Summary:**

In this paper, the authors propose to depict the dependencies crucial for modeling long sequences by bipartite mutual information. Though comprehensive experiments, they establish a power-law scaling of bipartite mutual information across diverse natural language datasets using LLMs. With this observation, they introduce the concept of long-context language modeling condition and prove that the model's state size should outpace the scaling of bipartite mutual information to achieve effective long-context modeling. This finding is verified on both transformer and state-space models.

**Questions:**

1. In line 258, the authors claim that bipartite mutual information reaches its maximum for a given $L$ when $\ell=L/2$. It is encouraged to provide more details to explain this result.
2. Estimating mutual information is challenging, particularly for high dimension random variables. What is the error of the mutual information estimation method used in this paper? Does it increase with dimensionality?

**Ethical Concerns:**

["NO or VERY MINOR ethics concerns only"]

**Final Justification:**

The authors have adequately addressed my concerns in the rebuttal. From my perspective, this paper is an interesting and solid work. I fully agree to accept this paper.

**Limitations:**

The authors have adequately addressed the limitations and potential negative societal impact of their work in a separate section after conclusion.

**Paper Formatting Concerns:**

I do not find any formatting issues in this paper.

**Quality:**

3

**Strengths And Weaknesses:**

**Strengths**
- The problem studied in this paper is significant and central, that is, the neural scaling law of LLMs. By conducting comprehensive experiments on recent state-of-the art LLMs, the authors establish a power-law scaling of bipartite mutual information for the first time, which is crucial to depict the long-range dependencies in natural language.
- Based on the above observation, the authors further theoretically demonstrate that the history state dimension must grow at least as fast as the power-law scaling of bipartite mutual information in the data, which brings new insights for community to design or improve the network architecture of LLMs.

**Weakness**
- Overall, this is an interesting and solid work from my perspective. The possible weakness could be that the authors only analyze the model's history state size but do not establish a relationship between bipartite mutual information and the model's expressive power or generalization capability. This could be a potential direction for future research.

---

> ### Author Rebuttal · Authors · 2025-07-31
>
> >**Strengths And Weaknesses:**
>
> >**Strengths**
>
> >-   The problem studied in this paper is significant and central, that is, the neural scaling law of LLMs. By conducting comprehensive experiments on recent state-of-the art LLMs, the authors establish a power-law scaling of bipartite mutual information for the first time, which is crucial to depict the long-range dependencies in natural language.
> >-   Based on the above observation, the authors further theoretically demonstrate that the history state dimension must grow at least as fast as the power-law scaling of bipartite mutual information in the data, which brings new insights for community to design or improve the network architecture of LLMs.
>
> We are grateful for the reviewer's assessment that our work addresses a significant and central problem in understanding neural scaling laws. Thank you for recognizing the importance of establishing power-law scaling of bipartite mutual information and its implications for architecture design.
>
> >**Weakness**
>
> >-   Overall, this is an interesting and solid work from my perspective. The possible weakness could be that the authors only analyze the model's history state size but do not establish a relationship between bipartite mutual information and the model's expressive power or generalization capability. This could be a potential direction for future research.
>
> We thank the reviewer for noting this potential future direction. Our work on data-driven information scaling and existing work on model-centric scaling laws (e.g., relating model size to generalization) are complementary. The former establishes a necessary condition on model state based on the data's intrinsic structure, while the latter studies how model capacity affects downstream performance. We agree that bridging these two perspectives to create a unified theory is a valuable and important direction for future research.
>
> We will add this to our discussion of limitations and future work, acknowledging that connecting information-theoretic scaling to model capabilities remains an open challenge.
>
> >**Questions:**
>
> >1.  In line 258, the authors claim that bipartite mutual information reaches its maximum for a given L when ℓ = L/2. It is encouraged to provide more details to explain this result.
>
> We thank the reviewer for this question. Bipartite MI often tends to be maximized near balanced partitions because it generally depends more strongly on the shorter side of the split ($I(X;Y) \le \min(H(X), H(Y))$). However, the precise location of the maximum can vary. In our appendix, we show results for multiple $\ell/L$ ratios demonstrating that the scaling law itself is robust to this choice.
>
> We will revise the text to say "tends to maximize" rather than "reaches its maximum" and add a brief explanation that bipartite MI generally depends on the shorter partition length.
>
> >2.  Estimating mutual information is challenging, particularly for high dimension random variables. What is the error of the mutual information estimation method used in this paper? Does it increase with dimensionality?
>
> We thank the reviewer for this important question. We acknowledge that estimating mutual information in high-dimensional settings is inherently challenging. Alternative methods like k-NN or neural estimators (MINE, InfoNCE) suffer from severe issues with the high and growing dimensionality of long text sequences. Our chosen methods, while not perfect, were the most robust and practical for this task, particularly as they do not exhibit rapidly increasing error with sequence length.
>
> While we provide preliminary error analysis in Appendix A, a complete quantification of the error is difficult, as it depends on how well the LLMs used in the estimation process capture the true data distribution. However, the clear and consistent power-law behavior we observe across different estimators and models gives us confidence in our main qualitative finding. We agree that refining the precision of the estimated exponent is a valuable direction for future work.
>
> >**Rating:**  5: Accept: Technically solid paper, with high impact on at least one sub-area of AI or moderate-to-high impact on more than one area of AI, with good-to-excellent evaluation, resources, reproducibility, and no unaddressed ethical considerations.
>
> We thank you again for your constructive feedback. We hope our responses and planned revisions have fully addressed your concerns. We would be happy to answer any further questions and kindly ask that you consider these clarifications in your final evaluation.

---

### Official Review · Reviewer_rSLc · 2025-07-01

**Clarity:** 3
**Significance:** 3
**Originality:** 3
**Rating:** 4
**Confidence:** 4

**Summary:**

The authors study the long-range dependencies of natural language for large language models. More precisely, they establish bipartite mutual information scaling laws (as power laws) that explain multi-token interactions beyond the prior understanding given by two-point mutual information. They explain how to estimate these scaling laws with large language models and demonstrate their validity with GPT2 and Mamba(2) models on the PG-19 dataset. The authors use these scaling laws to provide a condition, the $\text{L}^2\text{M}$ condition, that defines the minimum size of the latent space of a model to be able to deal with long-context dependencies. They validate this condition experimentally with GPT-2 and Mamba(2) models on synthetic data, showing that transformer-based models naturally satisfy the $\text{L}^2\text{M}$ condition while SSM need to scale their latent space size with the sequence length to be able to satisfy it.

**Questions:**

1) In the proof of Thm. 5.5, line 281, shouldn't it be "$I^{BP,q}$" instead of "$I^{BP}$"? Otherwise, it leads to comparing the same quantity in front and after the inequality sign.

**Ethical Concerns:**

["NO or VERY MINOR ethics concerns only"]

**Final Justification:**

The current work is interesting, and the authors addressed my concerns during rebuttal, except one related to the practical impact of the paper (this is not an experimental work but more in terms of how it could help design better LLMs using the current submission). I am not convinced how the current work could enable scaling up to millions or billions of tokens (which was claimed by the authors to explain the potential impact of their work), and I think that this claim is not supported in the current submission (even for future work building upon your work). The practical benefits of the current work are also unclear to other reviewers. As such, I will maintain my score (which leans towards acceptance).

**Limitations:**

Yes

**Quality:**

3

**Strengths And Weaknesses:**

*Strengths*
- The i sdeaeems novel, the paper is well written, and the problem tackled is well motivated
- I find the theoretical arguments compelling
- The experiments are comprehensive
- The connection between theory and experiments is done properly

*Weaknesses*

I list below what I believe are weaknesses, but I would be happy to be corrected if I misunderstood some important parts of the current submission.
1) The takeaway of the paper is not clear to me: indeed, transformer-based models always validate the LM condition because the key-value pairs scale linearly with the sequence length. Given that current SOTA models are transformer-based ones, how does the current paper improve our understanding of such models?
2) Related to the point above, to me, the author's proposal rather highlights the current limitations of SSM-based LLMs, whose sizes need to scale with the sequence length to be able to model long-context dependencies. As such, it is not clear how useful the proposed analysis and the LM condition are beyond "SSMs might suffer more from long-context". Could the author please elaborate on that?

*Reason for my current score*: Overall, I find the paper interesting and novel, with a well-conducted theoretical study with empirical verifications. However, the main message and the impact of the paper for the design, study, or use of LLMs are not clear to me. Notably, long context is indeed an issue, but we have seen increasing context windows in the recent open-source LLMs like the Llama, Gemma, Qwen, or Deepseek families. How does the proposed approach improve long-context abilities or our understanding of them beyond state-space models? These are the reasons why I lean towards a borderline acceptance, but I am willing to change my score provided the authors correctly address my concerns.

---

> ### Author Rebuttal · Authors · 2025-07-31
>
> >**Strengths And Weaknesses:**
>
> >_Strengths_
>
> >-   The idea seems novel, the paper is well written, and the problem tackled is well motivated
> >-   I find the theoretical arguments compelling
> >-   The experiments are comprehensive
> >-   The connection between theory and experiments is done properly
>
> We sincerely thank the reviewer for finding our work novel, well-written, and properly motivated. We appreciate the recognition of our compelling theoretical arguments and comprehensive experiments.
>
> >_Weaknesses_
>
> >I list below what I believe are weaknesses, but I would be happy to be corrected if I misunderstood some important parts of the current submission.
>
> >1.  The takeaway of the paper is not clear to me: indeed, transformer-based models always validate the L²M condition because the key-value pairs scale linearly with the sequence length. Given that current SOTA models are transformer-based ones, how does the current paper improve our understanding of such models?
>
> >2.  Related to the point above, to me, the author's proposal rather highlights the current limitations of SSM-based LLMs, whose sizes need to scale with the sequence length to be able to model long-context dependencies. As such, it is not clear how useful the proposed analysis and the L²M condition are beyond "SSMs might suffer more from long-context". Could the author please elaborate on that?
>
> We thank the reviewer for these important questions that allow us to clarify our contributions. Our main contribution is not simply to observe that transformers satisfy the L²M condition, but to establish L²M as a **necessary and universal lower bound** on latent state growth for *any* architecture to model the long-range dependencies observed in natural language.
>
> This perspective provides two key insights. First, it helps to reframe the challenges faced by architectures like SSMs. Our work suggests their struggle with long contexts is less about specific implementation details and more about a necessary architectural property we identify. The L²M condition requires the latent state to scale as $L^\beta$, posing a clear **design challenge** for any architecture, such as standard SSMs, that aims to use a fixed-size state. Second, and more importantly, it reveals that transformers, with their linearly growing key-value caches, actually **over-provision** latent state, since our empirical results show that only a sublinear growth ($L^\beta$ with $\beta < 1$) is required for natural language.
>
> This gap between the *necessary* sublinear scaling and the transformer's *actual* linear scaling opens an exciting avenue for future research: designing novel architectures (which could include sparse transformers or hierarchical transformers) that are more efficient than vanilla transformers (i.e., with sub-quadratic computational cost) while still precisely meeting the L²M requirement. Our work provides a concrete, data-driven target for this architectural innovation.
>
> We will revise the introduction and conclusion to better emphasize these implications for future architecture design beyond just explaining current limitations.
>
> >**Questions:**
>
> >1.  In the proof of Thm. 5.5, line 281, shouldn't it be "" instead of ""? Otherwise, it leads to comparing the same quantity in front and after the inequality sign.
>
> The reviewer is correct. We thank the reviewer for catching this error and will correct it in the updated version.
>
> >**Rating:**  4: Borderline accept: Technically solid paper where reasons to accept outweigh reasons to reject, e.g., limited evaluation. Please use sparingly.
>
> We thank you again for your constructive feedback. We hope our responses and planned revisions have fully addressed your concerns. We would be happy to answer any further questions and kindly ask that you consider these clarifications in your final evaluation.

---

> ### Comment · Reviewer_rSLc · 2025-08-01
> **Thanks You!**
>
> Dear authors,
>
> Thank you for your answers to my review. I better understand the takeaway from the paper, and I agree that the current submission will be strengthened by explaining it better. I also find the point of view on transformer linear scaling interesting.
> That being said, since modern LLMs seem to go towards increasing context window sizes, I wonder how the current condition can help design novel architectures and whether it is needed (e.g., sparse transformers have already been proposed in the literature, and modern LLMs adopt hardware and optimization solutions to solve the inference time and the context window scaling). Could the authors elaborate on these points?
>
> This question will not affect my recommendation, but do the authors have an idea of what kind of architecture could go beyond the linear scaling ($\beta < 1$)?
>
> [1] Child et al., Generating Long Sequences with Sparse Transformers, 2019

---

> ### Author Response · Authors · 2025-08-02
>
> Dear Reviewer rSLc,
>
> Thanks for getting back to us. As you've noted, increasing context window sizes is a very important direction right now, and our work aims to understand what's needed for a model to handle long context lengths. Increasing context window length is difficult for vanilla transformers due to the quadratic complexity. While sparse transformers have been proposed, there has been very little understanding of what level of sparsity is ideal for retaining the ability to model long context lengths while having minimum cost, which is key for designing architectures for long or infinite context windows. Our work bridges this gap and points out that we should aim for minimum-cost architectures that meet the L²M condition.
>
> As far as we know, such an architecture doesn't exist yet: architectures are often either too sparse (logarithmic scaling/constant scaling) or too dense (linear scaling), which further highlights how our work provides the guiding principle for long context lengths.
>
> We hope we've answered your questions. Given your positive assessment of our work's novelty and theoretical contributions, we would appreciate if you could consider raising the score given these further clarifications.

---

> > ### Comment · Reviewer_rSLc · 2025-08-05
> >
> > I thank the authors for answering my last questions.
> >
> > This does not fully convince me, given that an increasing number of models and very large ones manage to deal with a very large context window (> 128K) without relying on sparse transformers. As such, I am not convinced that the current work would lead to improvement in this direction (although it is not a requirement for acceptance because authors cannot predict the future, this follows my original weakness regarding the impact of the paper). I will maintain my score but remain open to slightly increasing it depending on discussions with other reviewers.
> >
> > Thanks again for your answers.

---

> > > ### Author Response · Authors · 2025-08-06
> > >
> > > Thanks again for the valuable comments and for recommending acceptance of our work!
> > >
> > > We just wanted to further clarify that while current LLMs can have very long context windows, they still suffer from inherent quadratic computational complexity, making it difficult to scale up further. This becomes critical for extended reasoning or coding tasks. When dealing with even longer contexts, people have to resort to compressing past information (e.g., https://arxiv.org/abs/2404.07143v1) to avoid this quadratic complexity. Without a principled understanding of how mutual information scales, past information can be either over-compressed (losing important information) or under-compressed (wasting computational resources and memory). Our work provides this missing principle, which we believe will be valuable for designing efficient LLM architectures that can handle millions or billions of tokens.
> > >
> > > We hope this further clarification could resolve the reviewer's concerns!

---

> > > > ### Comment · Reviewer_rSLc · 2025-08-08
> > > > **Final rating**
> > > >
> > > > I thank the authors for the additional clarifications.
> > > >
> > > > Again, I am not convinced how the current work could enable scaling up to millions or billions of tokens, and I think that this claim is not supported (even for future work building upon your work). The practical benefits of the current work is also unclear for other reviewers. As such, I will maintain my score (which leans towards acceptance).
> > > >
> > > > I thank the authors again for the good work and for answering my questions.

---

### Official Review · Reviewer_cZLb · 2025-07-02

**Clarity:** 3
**Significance:** 2
**Originality:** 3
**Rating:** 4
**Confidence:** 2

**Summary:**

This paper proposes and empirically substantiates a scaling law for bipartite mutual information in natural language, arguing that such a measure more comprehensively captures the long-range dependencies critical for long-context language modeling than traditional two-point (token-to-token) mutual information. Using empirical analyses with LLMs and diverse datasets, the authors demonstrate that bipartite mutual information scales as a power law. From this, they formalize the "Long-context Language Modeling (L²M) condition," which prescribes that the latent state size of a model must scale at least as rapidly as the bipartite mutual information with respect to context length to effectively store predictive information. Theoretical results and extensive empirical studies are provided to validate these claims and elucidate architecture-specific implications for transformers and state-space models.

**Questions:**

* Can the framework be generalized or adapted for hybrid models (hybrid transformer and state-space model) or even other generative paradigms (e.g., diffusion models), or will fundamentally new information scaling laws be needed in those domains?
* Are there empirical indications showing that architectures meeting the L²M condition perform better in practical long-context tasks (e.g., document QA) compared to those that violate it? What might be the gaps between theoretical MI scaling capability and actual task performance?

**Ethical Concerns:**

["NO or VERY MINOR ethics concerns only"]

**Final Justification:**

Overall, I believe this paper can provide a valuable theoretical supplement to the field of long context modeling. However, its integration with the downstream performance of modern PLMs with large context window in real-world scenarios remains insufficient, and thus it cannot offer direct insights for now. Therefore, I will maintain my score (borderline accept).

**Limitations:**

yes

**Quality:**

3

**Strengths And Weaknesses:**

**Strengths**
* Clear Theoretical Innovation: The manuscript introduces a rigorous, information-theoretically grounded scaling law for bipartite mutual information in natural language sequences. By establishing the distinction and practical importance of bipartite over two-point mutual information, the authors provide a novel lens through which to view long-range dependencies.

* Empirical Verification: The claims are validated both on synthetic datasets and on real-world data. The L²M condition can give actionable guidance for model designers, linking information scaling properties with minimal latent state requirements.


**Weaknesses**

* Bipartite MI Estimation: While two estimators (direct, vCLUB) are used, the paper acknowledges that both likely underestimate the true scaling exponent β, yet the quantitative effect remains unclear.  The reliability of exponent fits completely resolved.
* It appears that the appendix is missing.

---

> ### Author Rebuttal · Authors · 2025-07-31
>
> >**Strengths And Weaknesses:**
>
> >**Strengths**
>
> >-   Clear Theoretical Innovation: The manuscript introduces a rigorous, information-theoretically grounded scaling law for bipartite mutual information in natural language sequences. By establishing the distinction and practical importance of bipartite over two-point mutual information, the authors provide a novel lens through which to view long-range dependencies.
>
> >-   Empirical Verification: The claims are validated both on synthetic datasets and on real-world data. The L²M condition can give actionable guidance for model designers, linking information scaling properties with minimal latent state requirements.
>
> We deeply appreciate the reviewer's recognition of our theoretical innovation and empirical validation.
>
> >**Weaknesses**
>
> >-   Bipartite MI Estimation: While two estimators (direct, vCLUB) are used, the paper acknowledges that both likely underestimate the true scaling exponent β, yet the quantitative effect remains unclear. The reliability of exponent fits completely resolved.
>
> We acknowledge that both estimators likely underestimate the true scaling exponent $\beta$. While we tried many alternative methods, each had significant limitations in our high-dimensional, long-sequence setting.
>
> In particular, K-nearest neighbor (KNN) estimators, while asymptotically unbiased, struggle severely with the high dimensionality of our task. In addition, since KNN only works for continuous random variables, tokens must be embedded into latent vectors first, which creates a significant tradeoff between choosing higher embedding dimensions for better representation versus lower dimensions where KNN works better. Neural estimators like MINE and InfoNCE face great difficulty in training neural networks on the vast distribution of long-sequence natural language. We suspect training such neural estimators is no less difficult than training LLMs themselves, since they must learn good representations of natural language correlations to provide meaningful mutual information estimates. Additionally, all other methods we tested exibits rapidly increasing error as sequence length grows (due to increasing dimensionality), either from inherent issues with high dimensionality (KNN) or increased difficulty in training neural networks (MINE and InfoNCE).
>
> We ultimately found vCLUB suffers the least from these problems, particularly due to its ability to leverage existing LLMs as the variational distribution. The estimated mutual information also doesn't suffer from increased dimensionality as sequence length increases. To validate our results, we also proposed the direct estimator, and cross checked scalings using different LLMs, all of which achieved consistent results, demonstrating the effectiveness of our approach. While the methods aren't perfect, our reported measurements represent the best currently achievable, and are reasonably good for our purpose as a first meaningful attempt to measure the mutual information scaling using LLMs. We would be happy to learn any other suggestions for more advanced estimation methods to refine these measurements in future work.
>
> We will update the paper to include a more detailed discussion of the challenges in estimating the mutual information scaling, and further clarify the limitations of our and other methods to further contextualize our methodological approach within the broader literature on mutual information estimation.
>
> >-   It appears that the appendix is missing.
>
> We apologize for the confusion. The appendix was included in the supplementary materials. We will ensure it is properly integrated into the main submission in the updated version.
>
> >**Questions:**
>
> >-   Can the framework be generalized or adapted for hybrid models (hybrid transformer and state-space model) or even other generative paradigms (e.g., diffusion models), or will fundamentally new information scaling laws be needed in those domains?
>
> We thank the reviewer for the question. The L²M condition is architecture-agnostic. Hybrid transformer-state space models can satisfy the condition as long as their effective history state scales faster than $L^\beta$. For diffusion-based language models, while the notion of "history state" becomes less clear for the diffusion process itself, these models often still generate text autoregressively when producing extended long sequences. Our framework remains applicable at the scale where autoregressive generation is necessary.
>
> We will add a discussion about the applicability of our framework to hybrid architectures and other generative paradigms.
>
> >-   Are there empirical indications showing that architectures meeting the L²M condition perform better in practical long-context tasks (e.g., document QA) compared to those that violate it? What might be the gaps between theoretical MI scaling capability and actual task performance?
>
> Our current results suggest that satisfying the L²M condition correlates with stronger performance in long-context tasks. However, there remains an interesting gap between theoretical MI capability and actual downstream performance. This gap likely arises from optimization dynamics, inductive biases, and task-specific effects. Closing this gap through systematic evaluation on diverse long-context benchmarks would be valuable future work. In addition, while different architectures can meet the same L²M condition, the scaling exponent for different architectures can be different, which may also affect the downstream task performance, and is worth further exploration in future works.
>
> We will update the paper to add a paragraph discussing potential future works along this direction.
>
> >**Rating:**  4: Borderline accept: Technically solid paper where reasons to accept outweigh reasons to reject, e.g., limited evaluation. Please use sparingly.
>
> We thank you again for your constructive feedback. We hope our responses and planned revisions have fully addressed your concerns. We would be happy to answer any further questions and kindly ask that you consider these clarifications in your final evaluation.

---

> > ### Author Response · Authors · 2025-08-06
> >
> > Dear Reviewer cZLb,
> >
> > We hope our response has answered your questions and concerns! If you need any further clarification, we're happy to provide additional details. If we've addressed your points, we'd be grateful if you could consider raising the score.
> >
> > Thanks a lot!

---

> > ### Comment · Reviewer_cZLb · 2025-08-08
> >
> > Thanks to the authors for their detailed response. Overall, I believe this paper can provide a valuable theoretical supplement to the field of long context modeling. However, its integration with the downstream performance of modern PLMs with large context window in real-world scenarios remains insufficient, and thus it cannot offer direct insights for now. Therefore, I will maintain my score.

---

### Official Review · Reviewer_jrcZ · 2025-07-03

**Clarity:** 3
**Significance:** 3
**Originality:** 3
**Rating:** 4
**Confidence:** 3

**Summary:**

This paper is concerned with scaling laws for long-context language modeling and specifically develops a seemingly novel information measure that ends up having an empirical power-law characterization.  This is what they call the bipartite mutual information, and is the standard Shannon mutual information but over long disjoint blocks.  An implementable (suboptimal/biased) estimator for this quantity is developed through the use of LLMs.  A condition on the state space requirements for LLMs is developed using the scaling law, and used to analyze Transformers and state-space models.

**Questions:**

If one looked at some other corpora, e.g. proteins, DNA sequences, or computer code, would the approach in this paper help clarify the distinctions between the domains, e.g. why smaller models seem to be okay for biological sequences than natural language?


Authors say that theoretical understanding of neural scaling laws remains limited, but they might be interested in work such as Nayak and Varshney "An Information Theory of Compute-Optimal Size Scaling, Emergence, and Plateaus in Language Models" (NeurIPS 2024 workshop) that actually uses information-theoretic principles.  See also references thereto.  I would be curious to know if there are connections between scaling laws for language as in the present work and the information-theoretic explanation of neural scaling laws.

**Ethical Concerns:**

["NO or VERY MINOR ethics concerns only"]

**Final Justification:**

This is a good paper, and hope it is accepted.  The authors have addressed many of the comments of all the reviewers.

**Limitations:**

yes

**Paper Formatting Concerns:**

References don't seem to be capitalized correctly.

**Quality:**

3

**Strengths And Weaknesses:**

STRENGTHS
The empirical finding (assuming the bipartite mutual information is well-estimated) of the scaling law is quite intriguing, in some sense moreso for computational linguistics than for machine learning.

The implication of the predictability characterization of natural language on the state space requirements is insightful.

WEAKNESSES
I'm not convinced that the estimator for bipartite mutual information developed and used in this work is the best it could be. Appendix A.IV is an improvement on [91] but doesn't draw on the huge literature on mutual information estimation that includes unbiasedness and optimality results.

There does not seem to be much connection to the LLM-as-universal predictor literature, which might help clarify some things on the state space requirement derivation.  Indeed, the universal prediction literature in information theory is often concerned with state space size.  "Transformers Learn to Compress Variable-order Markov Chains in-Context" and "Transformers are Universal Predictors" are two papers I could find.

The fact there are really three versions of Theorem 5.2 is not mentioned in the main paper, this could be clarified.

---

> ### Author Rebuttal · Authors · 2025-07-31
>
> >**Strengths And Weaknesses:**
>
> >STRENGTHS The empirical finding (assuming the bipartite mutual information is well-estimated) of the scaling law is quite intriguing, in some sense moreso for computational linguistics than for machine learning.
>
> >The implication of the predictability characterization of natural language on the state space requirements is insightful.
>
> We thank the reviewer for recognizing that our bipartite mutual information scaling law is both intriguing and impactful for computational linguistics, and that the L²M condition offers valuable insight into the latent state requirements of language models.
>
> >WEAKNESSES I'm not convinced that the estimator for bipartite mutual information developed and used in this work is the best it could be. Appendix A.IV is an improvement on [91] but doesn't draw on the huge literature on mutual information estimation that includes unbiasedness and optimality results.
>
> We agree with the reviewer that our estimator is not perfect, including its tendency to slightly underestimate the exponent. We tried many methods for estimating mutual information scaling, but found most suffer from more severe issues.
>
> In particular, estimators like K-nearest neighbor (KNN), while asymptotically unbiased, struggle severely with the high dimensionality of our task. In addition, since KNN only works for continuous random variables, tokens must be embedded into latent vectors first, which creates a significant tradeoff between choosing higher embedding dimensions for better representation versus lower dimensions where KNN works better. Neural estimators like MINE and InfoNCE face great difficulty in training neural networks on the vast distribution of long-sequence natural language. We suspect training such neural estimators is no less difficult than training LLMs themselves, since they must learn good representations of natural language correlations to provide meaningful mutual information estimates. Additionally, all other methods we tested exibits rapidly increasing error as sequence length grows (due to increasing dimensionality), either from inherent issues with high dimensionality (KNN) or increased difficulty in training neural networks (MINE and InfoNCE).
>
> We ultimately found vCLUB suffers the least from these problems, particularly due to its ability to leverage existing LLMs as the variational distribution. The estimated mutual information also doesn't suffer from increased dimensionality as sequence length increases. To validate our results, we also proposed the direct estimator, and cross checked scalings using different LLMs, all of which achieved consistent results, demonstrating the effectiveness of our approach. While the methods aren't perfect, our reported measurements represent the best currently achievable, and are reasonably good for our purpose as a first meaningful attempt to measure the mutual information scaling using LLMs. We would be happy to learn any other suggestions for more advanced estimation methods to refine these measurements in future work.
>
> We will update the paper to include a more detailed discussion of alternative estimation methods we considered and their limitations, to further contextualize our methodological approach within the broader literature on mutual information estimation.
>
> >There does not seem to be much connection to the LLM-as-universal predictor literature, which might help clarify some things on the state space requirement derivation. Indeed, the universal prediction literature in information theory is often concerned with state space size. "Transformers Learn to Compress Variable-order Markov Chains in-Context" and "Transformers are Universal Predictors" are two papers I could find.
>
> There is indeed a real but complementary connection. Works like "Transformers Learn to Compress Variable-order Markov Chains in-Context" and "Transformers are Universal Predictors" demonstrate that transformers can, in principle, approximate arbitrary variable-order Markov processes, establishing their universality in prediction. Our contribution is orthogonal but complementary: we identify the minimal latent state growth rate required to capture the empirically observed bipartite mutual information scaling in natural language. While universal predictor results show what is possible in principle, our L²M condition specifies what is necessary in practice to match the scaling structure of natural language.
>
> We will include in the updated paper discussions on the connection to universal prediction literature and how our work complements these theoretical capabilities with empirical requirements.
>
> >The fact there are really three versions of Theorem 5.2 is not mentioned in the main paper, this could be clarified.
>
> We thank the reviewer for this suggestion. Theorem 5.2 indeed has three versions under different assumptions. We will update the main text to explicitly note this and will separate the variants with their assumptions into different theorems in the appendix for clarity.
>
> >**Questions:**
>
> >If one looked at some other corpora, e.g. proteins, DNA sequences, or computer code, would the approach in this paper help clarify the distinctions between the domains, e.g. why smaller models seem to be okay for biological sequences than natural language?
>
> Our method is not restricted to natural language and can, in principle, be applied to any sequential domain. If biological sequences like proteins or DNA exhibit smaller scaling exponents $\beta$, or exhibit logarithmic or even slower mutual information scaling, the L²M condition predicts correspondingly smaller latent state requirements, which may explain why smaller models suffice in practice. Conversely, domains like code, where $\beta$ is closer to natural language, may require larger latent states. These are interesting and meaningful future directions worth further exploration.
>
> We will add a brief discussion in the conclusion about the potential application of our framework to other sequential domains.
>
> >Authors say that theoretical understanding of neural scaling laws remains limited, but they might be interested in work such as Nayak and Varshney "An Information Theory of Compute-Optimal Size Scaling, Emergence, and Plateaus in Language Models" (NeurIPS 2024 workshop) that actually uses information-theoretic principles. See also references thereto. I would be curious to know if there are connections between scaling laws for language as in the present work and the information-theoretic explanation of neural scaling laws.
>
> We thank the reviewer for bringing up this paper. Their work explains compute-optimal scaling using information-theoretic principles and provides valuable insights into neural scaling laws. Our framework complements their approach: while they focus on how compute and model size must scale for optimal training, our L²M condition establishes how latent states must scale to satisfy the predictive information structure of the data.
>
> We will include discussion of this work and its connections to our framework in Section 2 and discuss how both approaches use information theory but address different aspects of scaling.
>
> >**Paper Formatting Concerns:**
>
> >References don't seem to be capitalized correctly.
>
> We thank the reviewer for noticing this and will fix it in the updated version.
>
> >**Rating:**  4: Borderline accept: Technically solid paper where reasons to accept outweigh reasons to reject, e.g., limited evaluation. Please use sparingly.
>
> We thank you again for your constructive feedback. We hope our responses and planned revisions have fully addressed your concerns. We would be happy to answer any further questions and kindly ask that you consider these clarifications in your final evaluation.

---

> > ### Author Response · Authors · 2025-08-06
> >
> > Dear Reviewer jrcZ,
> >
> > We hope our response has answered your questions and concerns! If you need any further clarification, we're happy to provide additional details. If we've addressed your points, we'd be grateful if you could consider raising the score.
> >
> > Thanks a lot!

---

> > ### Comment · Reviewer_jrcZ · 2025-08-06
> >
> > Thanks to the authors for their detailed rebuttal: if the paper is accepted, I believe the changes to the manuscript that have been proposed (bettercontextualizing in literature, adding details on alternatives to entropy estimation and why they may not work well, etc.) will only strengthen it, so I encourage them to do so.
> >
> > I was hoping the authors would run an experiment on, e.g. a biological sequence model/dataset during the rebuttal period.  I remain quite curious of what will emerge.  Maybe for the camera-ready, if accepted?

---

> > > ### Author Response · Authors · 2025-08-07
> > >
> > > Thanks a lot for the suggestions! We will definitely include the proposed changes in the updated version of the paper.
> > >
> > > We also find the mutual information scaling of biological sequences quite interesting. We are currently working in this direction,  but it needs careful setup to do it properly (e.g., choosing the right models and datasets, ensuring fair comparisons across domains, etc.) Given the scope, we think this could be a nice follow-up study where we can explore it thoroughly. We'd really appreciate it if the reviewer had any suggestions about the biological sequence models or datasets.

---

> > > > ### Comment · Reviewer_jrcZ · 2025-08-07
> > > >
> > > > Comparing protein language models with genomic language models would itself be interesting, to say nothing of comparing to natural language.
> > > >
> > > > Also, on your earlier query of estimating mutual information, I was just remembering an older paper in the information theory literature that is fairly different from other approaches you've considered and might be inspirational in some way: [Arnold, Loeliger, Vontobel, Kavcic, and Zeng, "Simulation-Based Computation of Information Rates for Channels With Memory," IEEE Trans. IT, 2006].

---

> > > > > ### Author Response · Authors · 2025-08-07
> > > > >
> > > > > Thank you for all the suggestions!

---

### Decision · Program_Chairs · 2025-09-17

**Decision:**

Accept (poster)

**Comment:**

All four reviews of the paper leaned towards acceptance with three recommendations for borderline acceptance [jrcZ,cZLb,rSLc] and one recommendation for acceptance [gdYy].

Reviewers appreciated several aspects of the work:

+ The idea was considered novel [rSLc] and the problem was considered well motivated [rSLc] and significant [gdYy].
- The work was seen to have clear theoretical innovation [cZLb] and the theoretical arguments were considered compelling [rSLc]
+ The empirical finding of the scaling law was considered intriguing [jrcZ] and bringing new insights [gdYy]
+ The predictability characterisation of natural language on state space requirements was considered insightful [jrcZ]
+ The empirical verification was appreciated [cZLb] and experiments were considered comprehensive [rSLc], and connection between theory and experiments was appreciated [rSLc]
+ The paper was considered well written [rSLc]


Hovewer, a number of downsides were noted by the reviewers, and authors responded to them in the rebuttal stage:

- The estimator used for bipartite mutual information was criticised as not making use of existing literature on mutual information estimation [jrcZ]; authors argued several estimators had issues and stated they would add discussion.
- The takeaway was considered unclear in terms of how the paper improves understanding of transformer-based models [rSLc], and it was unclear how useful the analysis and condition are beyond SSMs suffering with long context [rSLc]. Authors argued their main contribution is to have the L^2M condition as a necessary universal lower bound, and discussed the implications on design challenges and overprovisioning in transformers. Some further discussion arose on potentially going beyond linear scaling and very long context windows.
- Lack of a connection between bipartite mutual information and expressive power/generalization capability was considered a weakness [gdYy]; authors considered it a future direction but would add brief discussion.
- Lack of connection to LLM-as-universal-predictor literature was criticised [jrcZ]; authors briefly discussed a connection.
- The quantitative effect of underestimating the scaling exponent and reliability of the exponent fits was considered unclear [cZLb]; authors provided some discussion on the estimation challenges.
- A clarity issue with versions of Theorem 5.2 was raised [jrcZ]; authors stated they would clarify.
- A question arose on generalisation to hybrid models or other generative paradigms [cZLb]; authors briefly commented hybrid models could satisfy the condition in some cases and stated to add a discussion on applicability of the framework.
- Another question was raised on whether architectures meeting the condition perform better in long-context tasks [cZLb]; authors noted their results suggest yes but discussed a gap to actual downstream performance, stating they would discuss this as future work.
- A question arose about use of the approach to clarify differences between domains [jrcZ]; authors provided brief discussion.
- A question on relation to other related work arose [jrcZ]; authors briefly discussed their framework is complementary.
- Additional detail on the maximum of the bipartite mutual information was requested [gdYy]; authors provided discussion and would add clarification.
- Discussion on error of the mutual information estimation method was requested [gdYy]; authors acknowledged the estimation is challenging and that complete error quantification is difficult, but left better estimation of the exponent as future work.
- An appendix was said to be missing [cZLb] but this was simply in the supplementary material.


After the rebuttal one reviewer [jrcZ] further hoped for an experiment on a biological sequence model/dataset but authors wished to leave it for follow-up work. Reviewer [cZLb] noted integration with downstream performance remained insufficient. Reviewer [rSLc] felt all but one concern was addressed, but was not convinced the work could enable scaling to large numbers of tokens, and felt practical benefits were unclear. Reviewer [gdYy] felt their concerns were addressed.

Overall, it seems reviewers appreciated the work, and it seems to make a useful contribution to understanding scaling laws for language modelling. While some concerns were left for future work, it seems the work can be at a sufficiently good state to be presented at NeurIPS.